# MarsGT: Multi-omics analysis for rare population inference using single-cell graph transformer

Xiaoying Wang[1,2,3,6], Maoteng Duan[1,6], Jingxian Li[1], Anjun Ma [2,3], Gang Xin[3], Dong Xu[4,5], Zihai Li [3], Bingqiang Liu [1,7] ✉ & Qin Ma [2,3,7] ✉

Rare cell populations are key in neoplastic progression and therapeutic response, offering potential intervention targets. However, their computational identification and analysis often lag behind major cell types. To fill this gap, we introduce MarsGT: Multi-omics Analysis for Rare population inference using a Single-cell Graph Transformer. It identifies rare cell populations using a probability-based heterogeneous graph transformer on single-cell multi-omics data. MarsGT outperforms existing tools in identifying rare cells across 550 simulated and four real human datasets. In mouse retina data, it reveals unique subpopulations of rare bipolar cells and a Müller glia cell subpopulation. In human lymph node data, MarsGT detects an intermediate B cell population potentially acting as lymphoma precursors. In human melanoma data, it identifies a rare MAIT-like population impacted by a high IFN-I response and reveals the mechanism of immunotherapy. Hence, MarsGT offers biological insights and suggests potential strategies for early detection and therapeutic intervention of disease.

Multicellular organisms encompass a diverse range of specialized cells. Identifying these cell types is pivotal in immunotherapy and clinical scenarios, as it illuminates immune mechanisms, aids in devising targeted therapies, and bolsters personalized medicine by unmasking the unique cellular makeup of each patient[1]. However, difficulties surface when encountering rare or transiently expressed cells[2,3]. Despite their scarcity, rare cell populations step up to play crucial roles in a variety of biological processes[4,5]. For example, antigen-specific memory T cells are integral for sustained immunosurveillance and long-term immunity, even in infection-free periods[6]. Conversely, invariant natural killer T cells impact a variety of pathologies, including microbial infections and autoimmune diseases, due to their robust immunoregulatory functions[7,8]. Additionally, minimal residual disease, denoting the minute cancer cell population post-treatment, acts as a significant early

indicator for potential tumor relapse, highlighting the necessity to identify and comprehend these rare cell groups in disease dynamics and therapeutic interventions[9,10]. A refined grasp of these rare cell populations, culminating in a more detailed depiction, will illuminate our understanding of tumor microenvironments and the intricate mechanisms that steer the responses to immunotherapy.

The advent of single-cell RNA sequencing (scRNA-seq) has vastly improved our ability to identify individual cell types, offering high-resolution molecular profiles that illuminate cellular diversity and the complex dynamics of gene expression within specific cells[11,12]. Most existing rare cell identification tools, such as FIRE[4], GapClust[13], TooManyCells[14], GiniClust[15], RaceID[2], and SCMER[1], confront several challenges, such as high false positives when inferring rare populations, limited performance with complex samples like tumor biopsy

[1]School of Mathematics, Shandong University, Jinan, Shandong 250100, China. [2]Department of Biomedical Informatics, College of Medicine, The Ohio State University, Columbus, OH 43210, USA. [3]Pelotonia Institute for Immuno-Oncology, The James Comprehensive Cancer Center, The Ohio State University, Columbus, OH 43210, USA. [4]Department of Electrical Engineering and Computer Science, University of Missouri, Columbia, MO 65211, USA. [5]Christopher S. Bond Life Sciences Center, University of Missouri, Columbia, MO 65211, USA. [6]These authors contributed equally: Xiaoying Wang, Maoteng Duan. [7]These authors jointly supervised this work: Bingqiang Liu, Qin Ma. ✉e-mail: bingqiang@sdu.edu.cn; qin.ma@osumc.edu

single-cell data, inability to concurrently identify major and rare cell types, and compromised accuracy with ultra-rare cell types (<1% of the sample)[14]. These issues could stem from the limited representation of rare cells, which may lead to inaccurate grouping with more prevalent cell populations when solely relying on gene expression data. This pursuit can be further accelerated by technological innovations like single-cell ATAC sequencing (scATAC-seq)[12]. When synergistically used with scRNA-seq, these methodologies provide partial regulatory data concerning enhancer regions pivotal in preserving cell type identities[1]. This invaluable information can be tapped into for the construction of gene regulatory networks, thereby unraveling critical insights into the nature and function of rare cell populations[16].

Meanwhile, graph neural networks have recently demonstrated profound proficiency in deciphering complex biological data, offering robust backing for the precise analysis and study of scMulti-omics data[16–20]. The implementation of the heterogeneous graph transformer provides a unified framework that amalgamates diverse single-cell data types, thereby facilitating a comprehensive understanding of cellular heterogeneity[16,21–23]. This approach unveils the intricate interplay among various cell types within complex cellular landscapes, enhancing our comprehension of biological systems and bolstering opportunities for precision therapeutic interventions.

To fill the gap and validate the theory, we developed MarsGT (**M**ulti-omics **a**nalysis for **r**are population inference using **s**ingle-cell **G**raph **T**ransformer), an end-to-end deep learning model for rare cell population identification from scMulti-omics data. Graph neural networks have recently demonstrated profound proficiency in modeling single-cell data[24,25]. Furthermore, our in-house tool, DeepMAPS[16], has shown the superior performance of heterogeneous graph transformer (HGT), a powerful graph neural network architecture that can deal with large-scale heterogeneous and dynamic graphs, in biological network inference and cell clustering from the joint analysis of scMulti-omics data. With such a foundation, MarsGT introduces a probability-based HGT framework to analyze scMulti-omics data from a heterogeneous graph, including cells, genes, and peaks, which can build peak-gene regulatory relationships and utilize such relationships to characterize rare cell populations.

MarsGT, as a probability-based subgraph-sampling method, can highlight rare cell-related genes and peaks in a heterogeneous graph. We conducted extensive simulations ($n = 550$) to thoroughly test the accuracy and robustness of MarsGT in identifying rare cell populations. The performance of MarsGT, validated on the above simulation data and four human peripheral blood mononuclear cell datasets, surpassed existing methods in F1 score and Normalized Mutual Information (NMI) metrics. To further showcase the application capability of MarsGT, we applied MarsGT on three scMulti-omics case studies of (1) mouse retina, (2) human Fresh Frozen Lymph Node with lymphocytic lymphoma, and (3) melanoma patients and healthy donors. Our results demonstrate that MarsGT can distinguish unique rare cell populations—a feat not achievable with other computational tools—and provide strategies for early clinical detection and the development of immunological blockers.

## Results

### Overview of the MarsGT framework

MarsGT incorporates scRNA-seq and scATAC-seq data, and concurrently yields primary and rare cell populations along with their respective regulatory relations (Fig. 1 and Supplementary Fig. 1). A heterogeneous graph, comprising cells, genes, and enhancers, is constructed from the initial scRNA-seq and scATAC-seq data, with the presence of genes and peaks within cells represented as edges. We posit that a gene ubiquitously expressed is less likely to be pivotal for identifying rare cells compared to a gene that is expressed only within a specific subpopulation. To discern rare cells, it is imperative to identify genes or peaks that are highly expressed in a target cell but

exhibit low or no expression in other cells. We defined such genes and peaks as rare-related genes and peaks. The genes/peaks within a cell are segmented into high or low-selection regions according to the first quartile of the expression/accessibility. For a given cell, the selection probability of a gene/peak is determined by the proportion of gene/peak expression/accessibility in the high selection region. Such rare-related genes and peaks have a higher probability of being sampled to the key features of rare cells in our multi-head attention graph transformer. The multi-head attention mechanism facilitates the update of joint embeddings of cells, genes, and peaks on the sampled subgraphs. The cell assignment probability matrix and peak-gene link assignment probability matrix are predicted post-learning joint embedding. The peak-gene relations and rare cell populations from the subgraphs are concurrently determined and iteratively updated for model training. Furthermore, to safeguard that features pertinent to major cells are not diminished, regularization terms are incorporated into the training process (Supplementary Fig. 2). The fully trained model is subsequently applied to the entire heterogeneous graph, and a transcription factor (TF) database is incorporated to construct cell cluster enhancer gene regulatory networks (eGRNs)[12] (Methods).

### MarsGT achieves superior performances in rare and major cell population identification simultaneously on simulated and real data

We assessed the performance of MarsGT in identifying both rare and major cell populations across 550 simulated matched scRNA-seq and scATAC-seq datasets. To evaluate the performance of the tools on distinct datasets, we simulated 100 datasets using highly homogeneous cell line data. Each simulation dataset contained 500 cells and 2-3 cell types (Sim-CL 1, 2). To test these tools on heterogeneous data, we simulated an additional 300 datasets using peripheral blood mononuclear cell (PBMC) data. Each simulation dataset contained 500 cells and 2-3 cell types (Sim-PBMC 1, 2, 3, 4, 5, 6) (Supplementary Data 1, Methods). Furthermore, to ensure that the simulation datasets more closely resemble real data, we increased both the number of cell types and the number of cells. We generated 150 simulated datasets using peripheral blood mononuclear cells. Each simulation dataset contained 5000 cells and 5-15 cell types (Sim-PBMC 7, 8, 9) (Methods). Each simulated dataset included benchmark annotations from their original manuscripts. MarsGT was first compared with CellCUIS[26], FIRE[4], and GapClust[15], which operate as classification-like tools, to infer rare cells only. The performance was evaluated based on the F1 score, Precision, and Recall metrics for rare cell identification performance. To ensure fairness, each benchmarking tool was also tuned with different parameter combinations (Methods). We selected the parameter combination for performance comparison based on the grid search benchmarking of all the above tools. Specifically, if the mean score of a parameter combination achieves the highest across all datasets, we consider it the default parameter (Methods). MarsGT outperformed all classification-like tools across Sim-CL 1-2 and Sim-PBMC 1-5 simulated datasets (totaling 350) in terms of F1 score, Precision, and Recall (Supplementary Fig. 3, Supplementary Data 2, and Source Data 1). MarsGT also outperformed all classification-like tools on the Sim-PBMC 7-9 simulated datasets (totaling 150) (Fig. 2a, Supplementary Fig. 4, and Source Data 2, 3). Furthermore, to verify MarsGT's rate of false positives, we compared it with other tools using the Sim-PBMC 6 dataset (totaling 50), which lacks rare cells (Methods). The results confirmed that MarsGT does not force the identification of rare cell populations (Supplementary Fig. 5, Supplementary Data 3, and Source Data 4).

Furthermore, to evaluate MarsGT's ability to identify major and rare cell populations simultaneously, we compared it with three clustering-like tools (GiniClust[13], RaceID[2], and SCMER[1]) using NMI, Purity, and Entropy metrics. In all simulation datasets, MarsGT surpassed all clustering-like tools across all simulation datasets in terms of

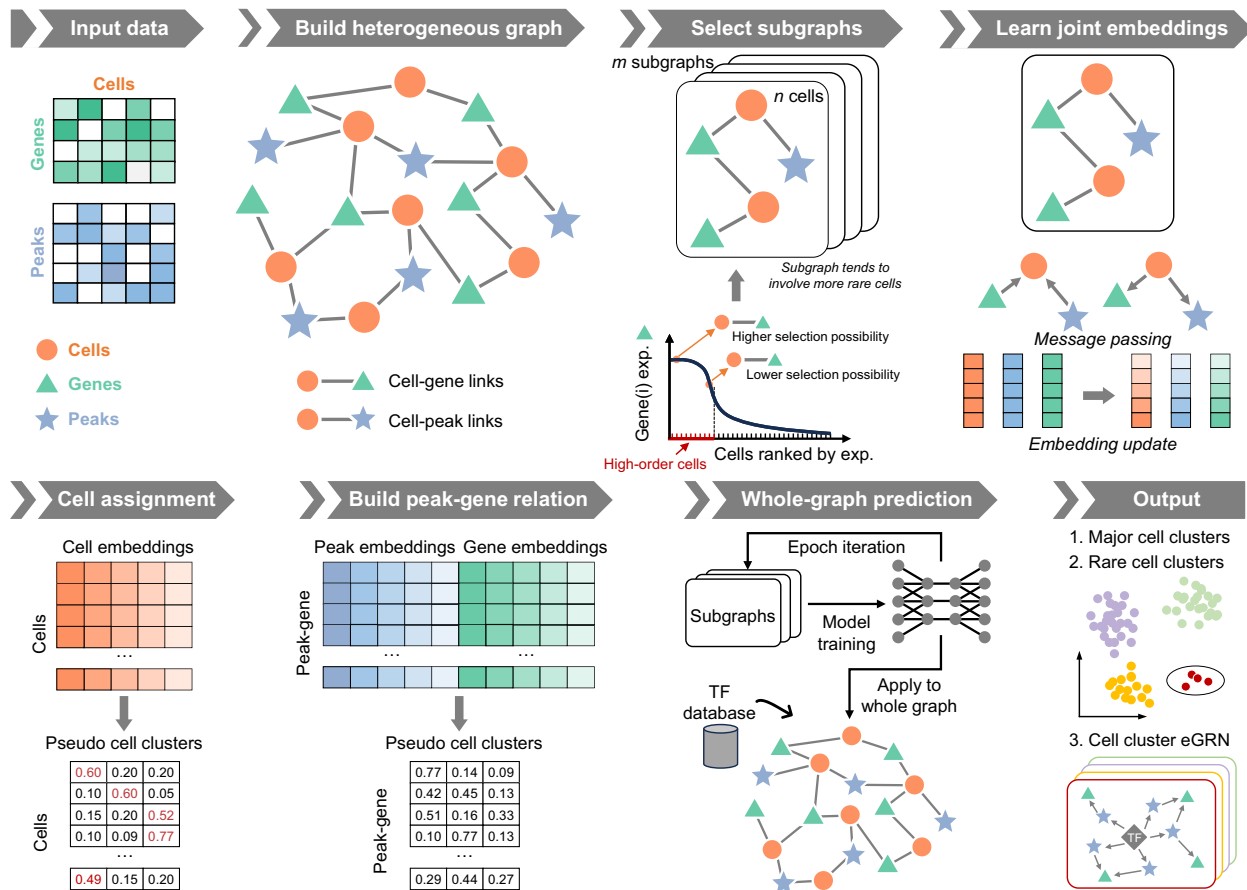

**Fig. 1 | The framework of MarsGT.** The model employs a six-step process for cell clustering and eGRN inference from matched scRNA-seq and scATAC-seq data. First, a heterogeneous graph, comprised of cells, genes, and peaks, is constructed from the original scRNA-seq and scATAC-seq data. Genes and peaks serving as edges are selected for a cell if they show high accessibility/expression within that cell and low accessibility/expression in others. The second step is learning joint embedding, which employs four graph relations to pass the message via a heterogeneous transformer. Arrows depict connections between target and source nodes. The third step involves cell assignment. Here, cell clusters are inferred from a probability matrix with rows representing cells and columns indicating pseudo-cell clusters. Cells that share the same maximum probability belong to the same cluster. The fourth step is constructing the peak-gene relationship via a matrix calculated from gene and peak embeddings, with rows denoting the regulatory potential of the peak to gene and columns indicating pseudo-cell clusters. In the final step, the trained model is applied to the entire graph. Following this, TF database information is integrated to infer cell clusters and eGRNs. Circles represent rare cell populations.

Purity and Entropy. Regarding the NMI index, GiniClust exhibited a similar performance to MarsGT (Fig. 2b, Supplementary Figs. 4, 6, Supplementary Data 2, and Source Data 2, 3, 5). A cell type is classified as a rare cell type if it constitutes less than 3% of the total cells. However, certain rare cells, such as senescent cells, can constitute an even smaller proportion[27]. To assess each tool's ability to identify these extremely rare cell types, we performed a gradient test with proportions of 0.5%, 1%, 2%, and 3% rare cells across five simulated datasets. A detailed comparison across all evaluation metrics demonstrated that MarsGT outperformed the existing top-performing tool in rare cell identification, exhibiting a superior F1 score that was 11.56%-143.49% higher, across different proportions of rare cells (Supplementary Fig. 7, Supplementary Data 4, and Source Data 6).

To evaluate MarsGT's performance on real datasets, we chose four datasets (PBMC-bench-1, 2, 3, and PBMC-test) from human peripheral blood mononuclear cells with ground truth labels. To maintain fairness, we presented the performance in a bar plot, using default parameters for all benchmarking tools and showing the results of parameter combinations (Supplementary Data 5, 6). In real data, we separately classified cell types constituting less than 3% and 1% of total cell counts as rare cell types. MarsGT achieved 100% and 63.71% higher F1 scores, compared to the second-best-performing tool (GiniClust), for 1% and 3% simulated rare cell proportion identification,

respectively in the independent test dataset (PBMC-test) (Fig. 2c). MarsGT delivered the best performance among all rare cell identification tools, achieving an NMI score that was 7.14% higher than that of the second-best-performing tool (GiniClust) in the independent test dataset (PBMC-test) (Fig. 2c and Source Data 7). The cell clustering UMAP on an independent dataset with benchmarking labels illustrated that MarsGT can accurately identify all rare cell types compared to other tools (Fig. 2d). The cell clustering UMAP on an independent dataset with benchmarking labels illustrated that MarsGT can accurately identify all rare cell types compared to other tools (Supplementary Fig. 8, Supplementary Data 7, and Source Data 8), which confirmed MarsGT's robust stability.

## MarsGT effectively captures differential regulatory mechanisms and uncovers biologically meaningful rare cell populations often missed by other tools

To underscore MarsGT's robust capability in identifying rare cell populations within species beyond humans, we utilized MarsGT on a published dataset involving matched single-nucleus RNA sequencing (snRNA-seq) and single-nucleus ATAC sequencing (snATAC-seq) performed on 9383 cells from the mouse retina (Supplementary Data 1). This study demonstrates MarsGT's capabilities in discerning major and numerous rare cell populations,

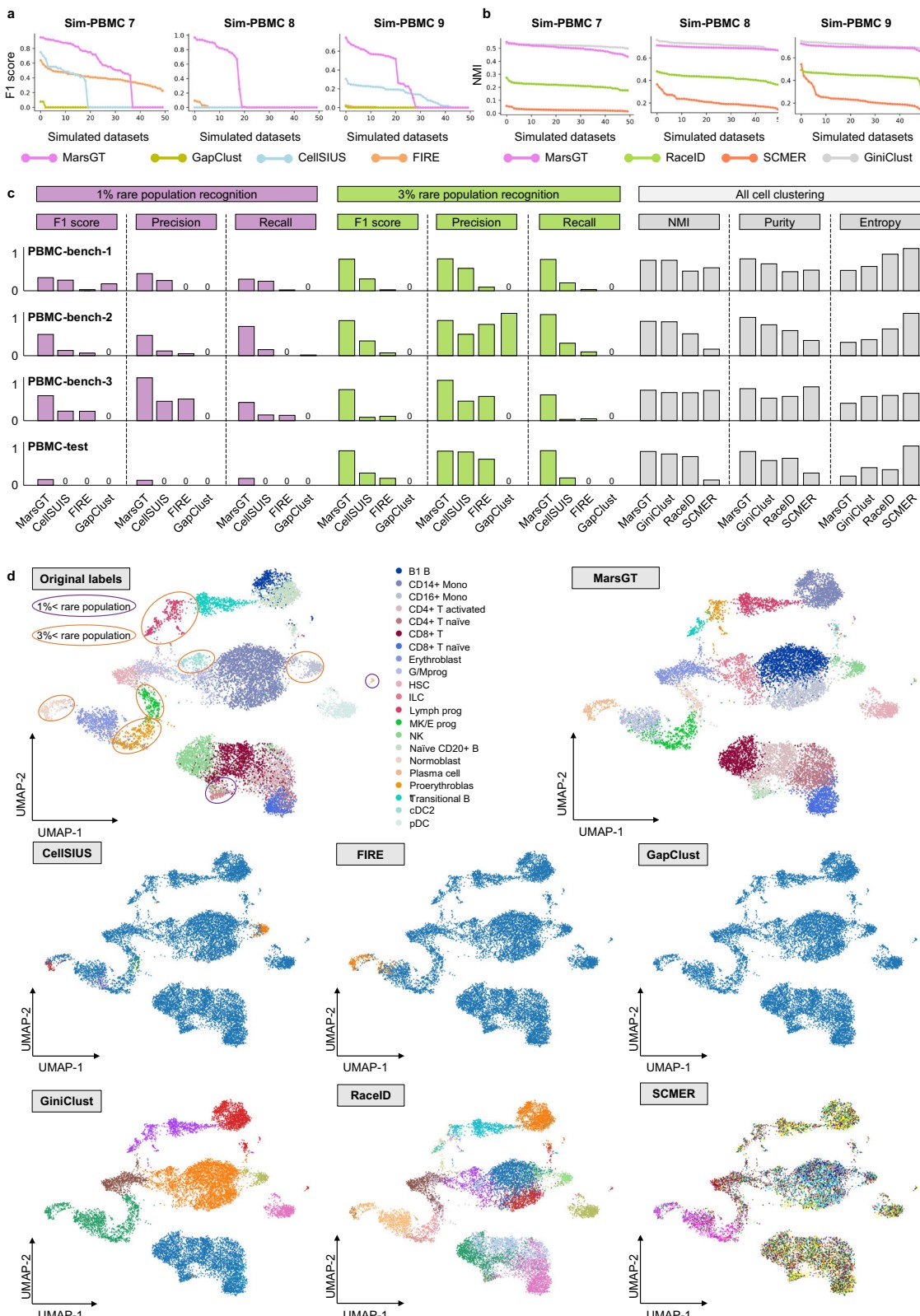

and we were able to identify 18 distinct cellular clusters, inclusive of one amacrine cell (AC) group, eight bipolar cell (BC) groups, one Cone cell group, one horizontal cell (HC) group, three müller glia cell (MG) groups, one retinal ganglion cell (RGC) group, and three Rod cell groups (Fig. 3a). Moreover, 12 rare cellular populations were distinguished, eight of which boast a 95% confidence level as highlighted by scPower[28] (Fig. 3b). The annotation of major cell

populations was accomplished through the visualization of expression levels pertaining to curated marker genes[29–31] (Fig. 3c). The populations of BC are known to exhibit a multitude of rare populations. Utilizing MarsGT, we identified eight unique populations of BC. These populations were annotated by visualizing the expression levels of curated marker genes specific to the BC subpopulation[32] (Fig. 3d). Excluding rod bipolar cell (RBC), all eight

**Fig. 2 | Benchmarking of MarsGT in terms of rare cell population identification.**
**a** Benchmark rare cell population identification on Sim-PBMC 7, 8, 9 datasets with classification-like tools in terms of F1 score. The X-axis signifies the dataset, while the Y-axis presents F1 scores arranged in descending order. **b** Rare cell population identification on Sim-PBMC 7, 8, 9 datasets benchmarked with clustering-like tools evaluated via NMI scores. The X-axis signifies the dataset, while the Y-axis denotes NMIs, organized in descending order. **c** Comparative results on three real training datasets (PBMC-bench-1, 2, 3) and one independent test dataset (PBMC-test). Test parameters across all tools are determined by the most optimal results obtained from the training dataset. **d** The UMAPS results for the independent PBMC-test dataset were calculated using PCA and predicted cell clusters in the tools. The purple and orange ellipses represent rare cell populations constituting less than 1% and 3%, respectively. Tools like FIRE and GapClust can distinguish only between major and rare populations in a binary fashion based on their method design. CellSIUS, due to its design, can identify several rare cell populations but cannot recognize major ones. Source data are provided as a Source Data file.

populations are regarded as rare cell populations at a 95% confidence level, as determined by scPower[28].

To validate whether the rare cell populations we identified is a false positive, further testing is required. Hence, we inferred potential cell-cell communications and constructed communication networks among different BC populations using CellChat[33]. Notably, we identified a non-canonical Wnt (ncWnt) signaling pathway originating from RBC and targeting both BC3 and BC6 (Supplementary Fig. 9, Supplementary Data 8). Previous research has highlighted the role of Wnt5a and Wnt5b, produced by RBC, in activating a non-canonical signaling pathway in rods, which in turn regulates early Outer Plexiform Layer (OPL) patterning[34], thereby validating the accuracy of the rare cell populations identified by MarsGT. We calculated the differentially expressed genes (DEGs) for each population further to elucidate the functionality of these distinct BC populations. Based on the cell population-specific DEG list, we inferred the functional pathways for each cell population (Fig. 3e). Notably, neuron migration was moderately enriched in BC1B, which aligns with its translocation from the bipolar to the amacrine cell layer. Categories such as axonogenesis and the glutamate receptor signaling pathway revealed modest differences among BC clusters. Interestingly, extracellular ligand-gated ion channel activity was predominantly enriched in OFF types, reflecting their employment of ionotropic glutamate receptors[35].

Interestingly, our analysis distinguished cluster 10, which comprises 127 cells, from cluster 2. Although both clusters are annotated as MG, cluster 10 stands out as a rare cell population. We denote cluster 2 as MG-1 and cluster 10 as MG-2. In the original paper, the 127 cells are annotated as Rod (120) and MG (5) by scRNA-seq and marker gene[36]. To further validate our findings, we utilized GiniClust[15], the algorithm with the second-best performance in our benchmarking section after MarsGT. GiniClust annotated the 127 cells as Rod (83 cells) and MG (41 cells) (Supplementary Fig. 10). This indicates that the rare MG identified in our study, which was not found in the original text, are not false positives. We ventured further into exploring the functional differences between the two MG clusters. Notably, we found MG-1 to be enriched in sprouting angiogenesis (Fig. 3f), suggestive of potential defects in retinal vascular development and a consequential functional deficit in MG, known to play a critical role in guiding outgrowing vessels[37]. MG-2 exhibited enrichment in the structural constituent of eye lens function (Fig. 3f). This finding echoes the assertions of previous research indicating that MG in both mature and embryonic retina binds antibodies generated to a lens fraction enriched for α-crystallin, a key lens protein[38]. The eGRN of the structural constituent of eye lens pathway related-gene is displayed in Supplementary Fig. 11 (Supplementary Data 9). We further visualized the eGRN for MG-1 and MG-2 (Fig. 3g, Supplementary Data 10, 11 and Source Data 9). Compared to other methods, MarsGT effectively captures differential regulatory networks, successfully identifying biologically meaningful rare cell populations that are often missed by RNA-only or other tools.

To further assert that MarsGT does not overlook pertinent information in the data while identifying rare cell populations, we initially compute the differentially expressed genes of the predominant cell populations. We identified top DEGs such as *Arr3*, *Gnat2*, and *Pde6h*, which act as marker genes for the Cone. Additionally, *hsd7a* serves as the marker gene for HC, whereas *Apoe*, *Clu*, and *Slc1a* function as marker genes for MG. *Meg3* is identified as the marker gene for RGC,

and *Nrl* for Rod[32] (Supplementary Fig. 12, Supplementary Data 12). The results validate the accuracy of MarsGT in identifying the predominant cell populations. Utilizing the raw gene expression data and cell populations, we deduced potential cell-cell communications and subsequently constructed communication networks among different cell populations within multiple signaling pathways, facilitated by CellChat[33]. Notably, we discovered a VEGF signaling pathway extending from AC, MG, RGC, HC, MG, and Rod towards MG as the target (Supplementary Fig. 13, Supplementary Data 13), which is consistent with the previous research[30].

## MarsGT identifies a rare state, B lymphoma-state-1, which offers the potential in preventing B-lymphoma progression

To underscore MarsGT's robust capability in identifying rare cell populations within cancer data, we utilized a matched scRNA-seq and scATAC-seq dataset available on the 10X Genomics website (Supplementary Data 1). This data originated from 14,566 cells obtained from a flash-frozen intra-abdominal lymph node tumor in a patient diagnosed with diffuse small lymphocytic lymphoma of the lymph node. MarsGT identified 14 distinct cell clusters, which we annotated by visualizing the expression levels of curated marker genes (Fig. 4a, b). Notably, four of these clusters were annotated as B cells. To differentiate the subpopulations of B cells, we visualized the expression levels of both normal B marker genes and B lymphoma marker genes across the four subpopulations (Fig. 4c). We designated one subpopulation as normal B cell population and three as lymphoma cell populations: B lymphoma-state-1 (BLS1), B lymphoma-state-2 (BLS2), and B lymphoma-state-3 (BLS3). BLS1, a rare cell population, exhibits a 95% confidence level, as indicated by scPower[28]. Intriguingly, it was evident that B cell subpopulations annotated using solely RNA or ATAC data could not be effectively distinguished (Fig. 4c, Supplementary Fig. 14). We also applied other scMulti-omics tools to the dataset for comparative purposes. For instance, Seurat[39] only identified two B cell subpopulations with hard annotation by curated marker genes (Supplementary Fig. 15), while our in-house tool DeepMAPS identified only three B cell subpopulations[16]. However, neither tool successfully identified the rare cell cluster within the B cells.

A pseudotime analysis on the four B cell clusters (comprising the normal B cell populations and three B lymphoma cell populations), using slingshot, postulated a lineage whereby the rare cell population BLS1 originates from the normal B cell populations. It was inferred that BLS1 predates BLS2, which in turn predates BLS3 (Fig. 4d, e). To substantiate this proposed linear-like developmental trajectory, we computed gene signature scores[40] reflecting different functions (anti-apoptosis, metastatic, and PD-PDL1) across the four B cell populations (Fig. 4f, Supplementary Data 14, Supplementary Data 15 and Source Data 10). These results depict a progressive progression of B lymphoma from BLS1 to BLS3. Focusing on *PDL1*, a critical gene in the PD1-PDL1 pathway that is promoted by *STAT1* and *HIF1A*, we observed more regulatory relations and intensity of *STAT1* and *HIF1A* in BLS3 (Fig. 4g, h, Supplementary Data 16). This finding is in line with our inferred linear-like development tendency. Furthermore, *BCL2*, an oncogenic gene that plays an anti-apoptotic role in cancer and drives its progression[41-44], demonstrated an incrementally enhanced regulatory score from normal B cells to BLS3 (Fig. 4i, Supplementary Data 16), lending further credibility to

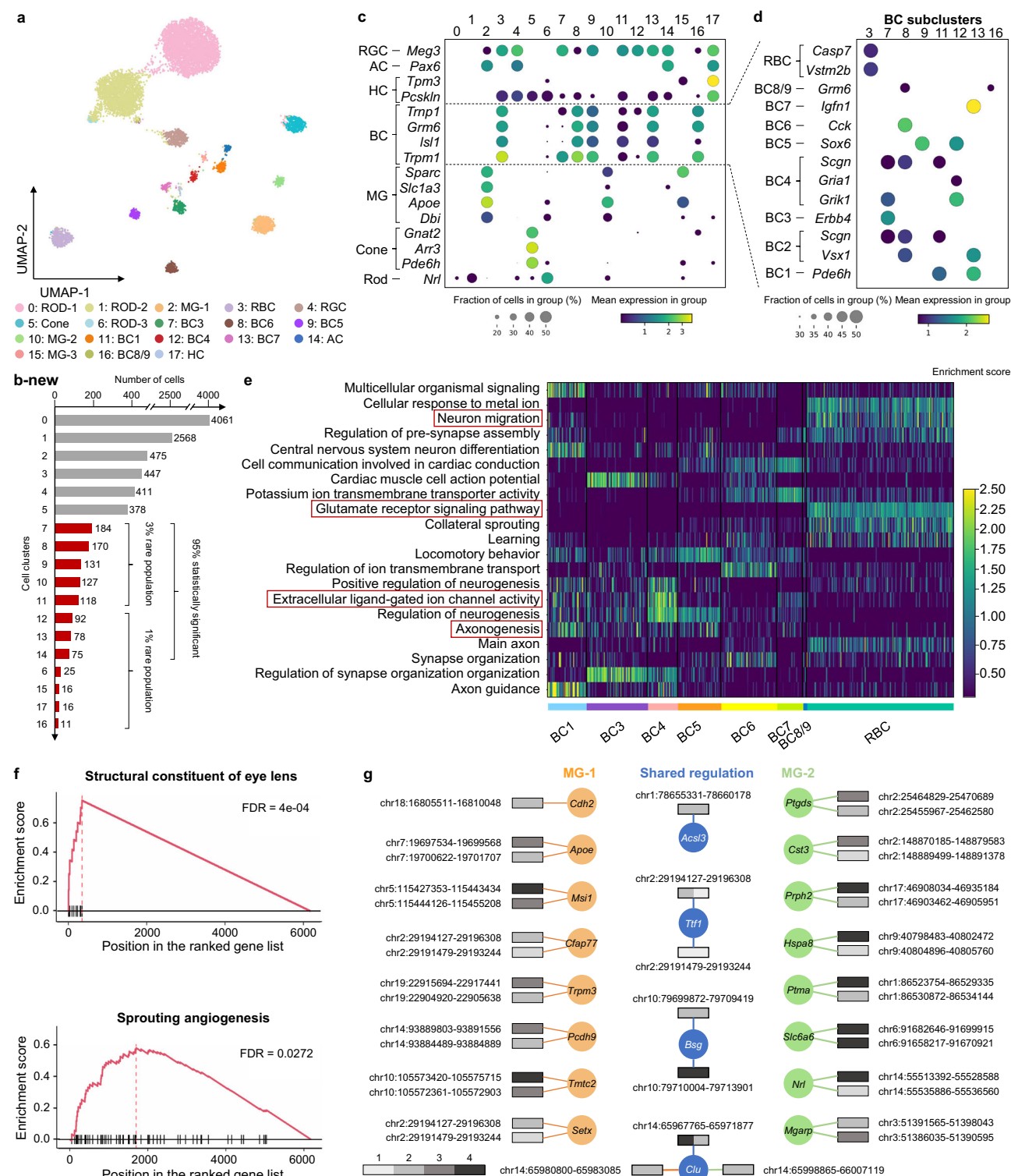

**Fig. 3 | MarsGT effectively captures differential regulatory mechanisms and uncovers biologically meaningful rare cell populations often missed by other tools. a** A UMAP visualizing the cell clusters predicted by MarsGT, annotated based on marker genes. **b** The number of cells in each cell cluster, with the red color signifying a 95% confidence level for rare cell populations. *p*-values were calculated based on two-tail negative binomial test. Multiple testing correction was performed by using FDR-adjusted *p*-values for scPower calculation. **c** Dotplot depicts the expression value and proportion of marker genes for the cell clusters predicted by MarsGT. **d** Dot plot represents the expression value and proportion of BC marker genes for the cell clusters predicted by MarsGT. **e** The enrichment score across each pathway is calculated using differentially expressed genes (DEGs) across various BC subpopulations. Pathways within the red box have been validated in the

literature. **f** Pathway enrichment as determined by Gene Set Enrichment Analysis (GSEA), based on the DEGs of MG-1 and MG-2. The structural constituent of the eye lens is the pathway enriched in MG-2, and Sprouting angiogenesis is the pathway enriched in MG-1. **g** The different peak-gene networks of MG-1 and MG-2. Rectangles symbolize the peaks, and their colors represent the mean accessibility of the peak in the cell population. Accessibility levels ranging from 0 to 0.1 are denoted as 1, those from 0.1 to 0.5 as 2, from 0.5 to 1 as 3, and anything above 1 is denoted as 4. The color of the line and circle represent the cell populations. The orange color signifies genes or relationships unique to MG-1, green indicates those unique to MG-2, while blue represents genes or relationships shared between MG-1 and MG-2.

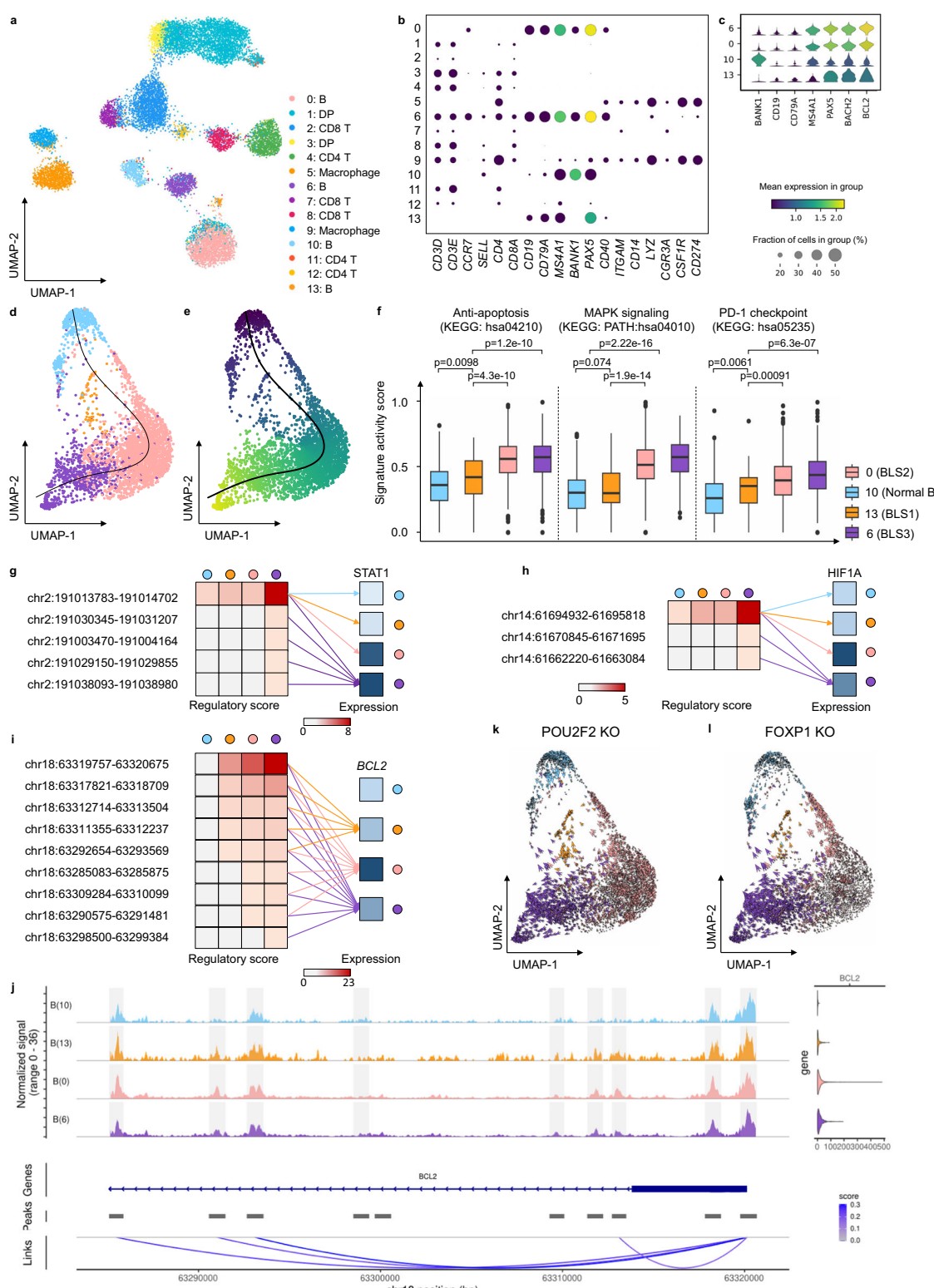

our inference. The scATAC-seq tracks of *STAT1*, *HIF1A* and *BCL2* are shown in Fig. 4j and Supplementary Fig. 16. Exploring BLS1 in greater depth, we compared the disparities among the four B cell clusters and identified a unique TF for BLS1, namely, MEF2C, along with switch-enhancer corresponding TFs: POU2F2, FOXP1, SPI1, and NFIC. Simulating a knockout experiment[45] involving these five TFs revealed a shift in B lymphoma cells towards normal B cells, suggesting that identifying the rare BLS1 state could offer potential avenues for curbing B-lymphoma progression (Fig. 4k, l,

Supplementary Fig. 17). It has been reported previously that MEF2C mutations lead to deregulated expression of the *BCL6* oncogene in B lymphoma[46,47], and that POU2F2 reflects the survival of B cell malignancies[48]. Additionally, FOXP1 is known to suppress immune response[49–52]. The roles of SPI1 and NFIC, as inferred by MarsGT, may provide fresh insights into therapeutic strategies for B lymphoma. In short, MarsGT can effectively identify a rare subset of B cells, BLS1, provides valuable insights into B-lymphoma progression, and opens new avenues for potential interventions, thereby advancing

**Fig. 4 | MarsGT identifies the rare cells in the intermediate transition state on B lymphoma data. a** UMAP visualizes cell clusters predicted by MarsGT, annotated based on the marker genes. **b** Dot plot demonstrates the expression value and proportion of marker genes within the cell clusters predicted by MarsGT. **c** A stacked violin plot represents the subpopulations of B cells, annotated with the marker genes for both normal and tumorous B cells. **d** A cell development trajectory for the four B cell subpopulations, with the line representing the lymphoma development trajectory. **e** The pseudotime of the four B cell subpopulations. **f** The gene signature scores of anti-apoptosis, metastatic, and PD-PDL1 pathways. Each box showcases the minimum, first quartile, median, third quartile, and maximum gene signature enrichment scores of different pathways in four cell clusters (Cluster 0: $n = 2754$, Cluster 6: $n = 731$, Cluster 10: $n = 456$, Cluster 13: $n = 72$). $p$-values were calculated based on two-tail $t$-test. Color represents cell clusters, and the Y-axis is the enrichment score. **g** The regulatory relationship of the PDL1 gene across the four B cell subpopulations. The red color signifies the regulatory score for each enhancer of the STAT1 coding gene across different B cell subpopulations, while blue indicates gene expression. **h** The regulatory relationship of the PDL1 gene across the four B cell subpopulations. The red color signifies the regulatory score for each enhancer of the HIF1A coding gene across different B cell subpopulations, while blue denotes gene expression across these subpopulations. **i** The regulatory relationship of the BCL2 gene (an anti-apoptosis promoting gene) across the four B cell subpopulations. The red color represents the regulatory score of each enhancer for the BCL2 coding gene across different B cell subpopulations, while the blue color represents gene expression in these subpopulations. **j** The Coverage plot for gene BCL2. The Coverage Plot encompasses the tracks of scATAC-seq (upper), peak links (lower), and gene expression (right). **k** Observed and extrapolated future states (depicted as arrows) following the POU2F2 knockout in the four B cell subpopulations. The color represents the different cell clusters. **l** Observed and extrapolated future states (depicted as arrows) following the FOXP1 knockout in the four B cell subpopulations. The color represents the different cell clusters. Source data are provided as a Source Data file.

---

our understanding of cellular disease dynamics and fostering innovative medical research and treatment strategies.

## MarsGT identifies MAIT-like rare cell populations and eGRNs in multi-sample melanoma scRNA-seq and scATAC-seq

To broaden MarsGT's scope in multi-sample and rare cell population inference, we utilized ten matched scRNA-seq and scATAC-seq samples of peripheral blood mononuclear cells (PBMCs) from melanoma patients at baseline (prior to receiving anti-PD1 therapy) and healthy donors. Of these, two samples were from healthy donors, while the remaining eight were from melanoma patients. MarsGT identified 13 distinct cell clusters from the integrated dataset of these ten samples, which include one CD4 + T cell group, three CD8 + T cell groups, two CD14+ monocytes (Mono) cell groups, one B cell group, one Nature killing (NK) cell group, one FCGR3A+ Mono cell group, and three Dendritic (DC) cell group. These clusters were annotated by visualizing the expression levels of curated marker genes in the major cell populations (Fig. 5a, b). DEGs for each cell population were calculated and represented in a heatmap (Supplementary Fig. 18). Notably, we observed two rare cell populations within the CD8 + T cells, namely, Cluster 9 and Cluster 12. The DEGs across these three CD8 + T cell populations were computed (Supplementary Fig. 19, Supplementary Data 17), leading to the identification of top DEGs such as ZBTB16 and SLC4A10 in Clusters 9 and 12. These genes act as markers for Mucosal Associated Invariant T cells (MAIT), recognized as a crucial rare cell population in immune responses. To delve deeper into the cell populations within Clusters 9 and 12, we visualized additional MAIT marker genes (Fig. 5c). Given that both MAIT and Natural Killer T (NKT) cells are non-canonical T cells and share similarities in their functions and partial marker genes, we computed the gene signature enrichment scores for MAIT and NKT respectively. The MAIT score was significantly higher than the NKT score (Fig. 5d, Supplementary Data 18 and Source Data 11), leading us to define Cluster 9 as MAIT-like 1 and Cluster 12 as MAIT-like 2.

We constructed the eGRN of the three CD8 + T cell populations and found that common enhancers and genes in the three cell populations constituted the majority proportion (Fig. 5e, f, Supplementary Data 19). Meanwhile, each cell population demonstrated unique enhancers and genes, indicating that MAIT-like 1 and MAIT-like 2 represent different MAIT-like subpopulations, each endowed with unique functional attributes. To support this observation, we inferred the pathways enriched in the MAIT-like cell population based on the cell population's active gene expression in the eGRNs (Fig. 5g). MAIT-like 1 and MAIT-like 2 shared several pathways, including Regulation of I-kappaB kinase/NF-kappaB Signaling, which play significant roles in immune responses. Unique to MAIT-like 1 were pathways such as Positive Regulation of Cytokine Production Involved in Immune Response, Interleukin-12-Mediated Signaling Pathway, and Regulation of Type I Interferon Production. The MAPK Cascade pathway was unique to MAIT-like 2, further differentiating these two subpopulations. Divergent regulatory patterns became apparent when we focused on a single regulon, ZBTB16, recognized as a critical transcription factor in MAIT cells. We visualized the expression of the genes it regulates, as well as the average accessibility value of the peaks associated with these regulated genes (Fig. 5h). We further mapped the regulatory relationships of ZBTB16 within the two MAIT-like cell populations (Fig. 5i). Interestingly, our results highlight minor differences in expression and accessibility, but more substantial variations in regulatory relationships. This supports our hypothesis that regulatory information is pivotal in recognizing rare cell populations.

## MarsGT reveals the mechanism for different survival of PD1-blocking immunotherapy

The above ten sample datasets we analyzed incorporated immunotherapy data from eight melanoma patients, grouped according to their Interferon-I response capacity (IRC). Four patients exhibited high IRC, as determined by the levels of Interferon-I stimulated proteins measured by mass cytometry, while the remaining four demonstrated low IRC (Supplementary Fig. 20). The source study inferred that a hyporesponsive IRC effectively predicted extended survival following PD1-blocking immunotherapy, while high responsiveness strongly associated with treatment failure and reduced survival duration. Interestingly, we observed that in both MAIT-like 1 and MAIT-like 2 cells, samples with low IRC represented a majority, accounting for 83.57% and 70.18%, respectively, a figure significantly higher than those with high IRC (Fig. 6a). Given the potential significance of MAIT-like 1 cells in understanding the survival mechanisms underpinning PD1-blocking immunotherapy, we decided to investigate this cell population further (MAIT-like 2, with only 69 cells, was excluded due to potential low confidence). We found 176 MAIT-like 1 cells in the high IRC group, compared to 895 cells in the low IRC group (Fig. 6b). Additionally, the count of DCs in the low IRC group exceeded those in the high IRC group. We compared the expression of IFN-I stimulated genes (ISGs) between high and low IRC patients.

Contrary to the IRC assignments, ISG expression in low IRC patients was significantly higher than in high IRC patients (Fig. 6c and Source Data 12). Upon calculating the unique enhancers in the eGRNs of high IRC and low IRC groups, respectively, we found that transcription factors TCF1 and BCL6 were exclusively present in the low IRC group. Previous studies have demonstrated that the TCF1-Bcl6 axis counteracts type I interferon to repress exhaustion and maintain T cell stemness[53]. Then we calculated the effector and exhaustion gene signature scores for MAIT cells in high IRC and low IRC groups, respectively (Fig. 6d, Supplementary Data 20 and Source Data 13). MAIT-like 1 cells in the high IRC group appeared exhausted, while those in the low IRC group appeared effective. This observation may explain the

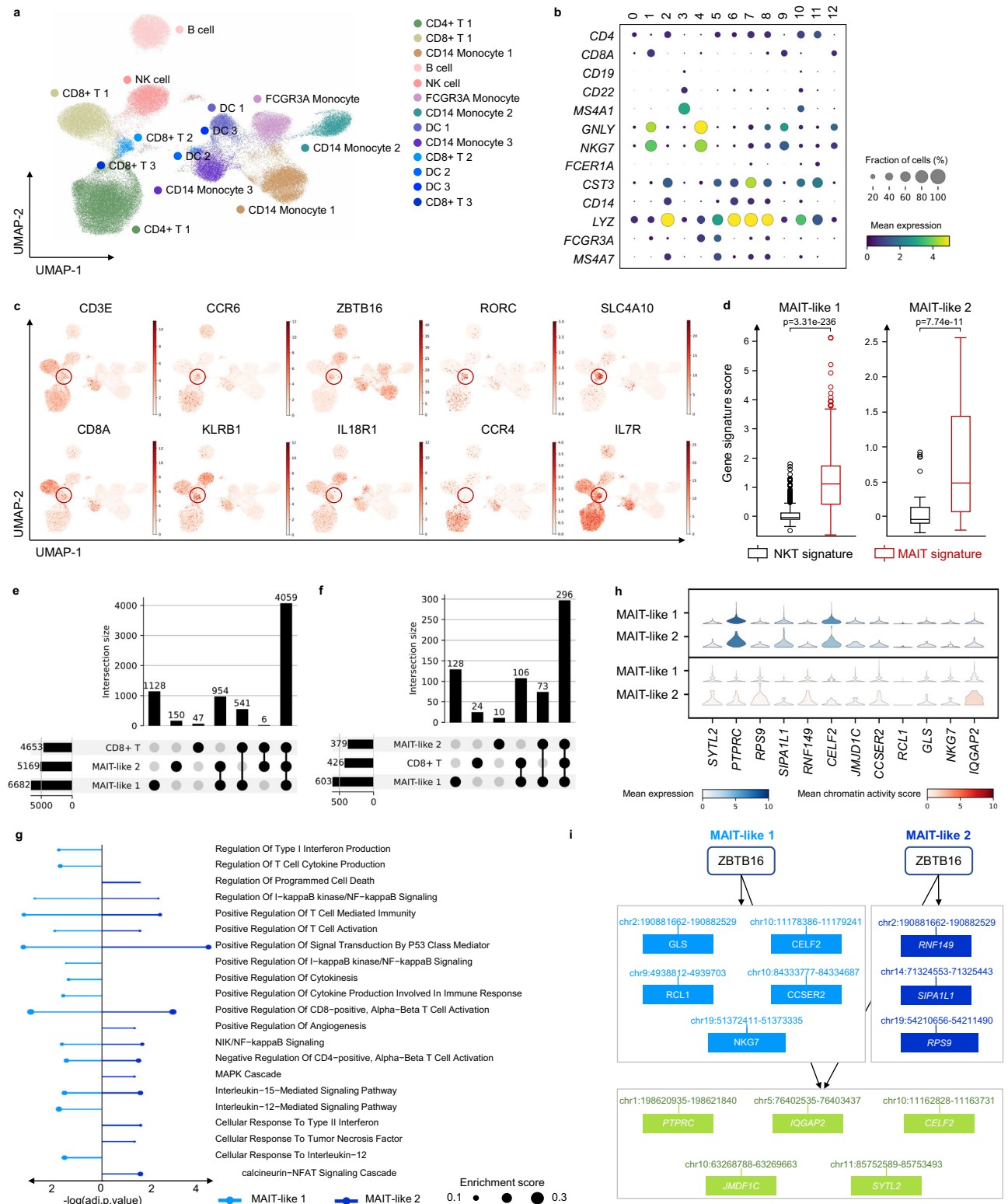

varying IRC across samples and the improved prognosis for low IRC patients following PD1-blocking immunotherapy.

High IRC suppressed MAIT-like 1 cells response by triggering high interleukin-10 (IL-10) production by DC, which subsequently inhibited the secretion of IL-15, IL-18 which are costimulatory cytokines for MAIT-like 1 cell activation. This observation is supported by the expression levels of the genes coding for these cytokines and their respective receptors (Fig. 6e–g and Source Data 14–16). The effector functions of MAIT-like 1 cells are mediated through the production of

IFN-II, GzmB, and Perforin (Fig. 6h). Our attention then turned to the regulatory mechanisms of IFN-II, GzmB, and Perforin in the high and low IRC contexts. In low IRC, IFN-I signaling is relayed by Tyk2-NFKB to regulate IFNG (the coding gene of IFN-II), and relayed by Jak1-STAT1 to regulate GZMB. IL-15 signaling is relayed by Jak1/Jak3-STAT1 to regulate GZMB, while IL-18 signaling is relayed by JNK-NFKB to regulate IFNG and by JNK-FOS/JUN to regulate GZMB and PRF1 (the coding gene of Perforin). In contrast, high IRC sees only IL-18 signaling relayed by JNK-FOS/JUN to regulate PRF1 (Fig. 6i). The complete GRNs of IFNG, GZMB,

**Fig. 5 | MarsGT identifies MAIT-like rare cell populations and eGRNs in multi-sample melanoma scRNA-seq and scATAC-seq. a** UMAP visualizes MarsGT's predicted cell cluster results, annotated based on marker genes. **b** Dotplot depicts the expression value and proportion of marker genes within the cell clusters predicted by MarsGT. **c** UMAPS showcases the marker genes of MAIT cells. **d** Showcasing the marker genes of MAIT cells. Each box showcases the minimum, first quartile, median, third quartile, and maximum gene signature enrichment scores on different MAIT-like cell populations (MAIT-like 1: $n = 1071$, MAIT-like 2: $n = 67$). The p-value is calculated by the Mann-Whitney U test with two-sided. **e** The upset plot illustrates the peaks present in CD8 + T cell subpopulations. **f** The upset plot demonstrates the genes found in CD8 + T cell subpopulations. **g** Pathway

enrichment across different MAIT-like cell populations. Colors represent distinct cell populations, while the size of the dots indicates the ratio of enriched genes. p-values were calculated based on one-sided hypergeometric test. Multiple testing correction was performed by using FDR-adjusted p-values. **h** The expression and accessibility of ZBTB16-regulated genes across various MAIT cell populations. **i** The regulatory relations of ZBTB16 in different MAIT-like cell populations. Green means the common regulatory relations between MAIT-like 1 and MAIT-like 2. Light blue means the regulatory relations unique in MAIT-like 1. Dark blue means the regulatory relations unique in MAIT-like 2. Source data are provided as a Source Data file.

and PRF1 are depicted in Supplementary Fig. 21. Interestingly, we observed a higher number of positive regulatory relations in low IRC compared to high IRC. This suggests a critical role for low IRC in facilitating the elimination of tumor cells. In other words, compared to samples with high IRC, those with low IRC stimulate the expression of cytokine and cytolytic molecule coding genes in CD8+ T cells via various costimulatory factors and pathways. This leads to the enhanced secretion of IFN-γ, TNF-α, Granzyme B, and Perforin, which in turn boosts the cytotoxic function of CD8+ T cells, promoting more effective tumor destruction.

## Discussion

MarsGT is an end-to-end deep learning model capable of inferring and identifying rare cell populations from scMulti-omics data using a heterogeneous graph transformer. To model and represent the scMulti-omics data, a heterogeneous graph is constructed, comprising nodes of cells, genes, and peaks. This configuration allows for the updating of joint embeddings of cells, genes, and peaks, facilitated by a multi-head attention mechanism. An end-to-end model benefits rare cell identification as it integrates scMulti-omics data with minimal information loss. The graph transformer can enhance the signal-to-noise ratio, addressing the high dropout feature of single-cell technologies, which in turn reduces the false positive rate in rare cell identification. Crucially, MarsGT employs a probability-based subgraph-sampling technique during model training, which allows the selective highlighting of cell-gene and cell-peak relationships that are relevant to rare cells. In parallel, the model determines peak-gene relations and rare cell populations from the subgraphs, undergoing iterative updates during the training process. Consequently, MarsGT is well equipped to identify rare cell populations and their corresponding gene regulatory networks within the entire heterogeneous graph.

MarsGT's performance remains consistent across data types (snRNA-ATAC-seq or scRNA-ATAC-seq), health statuses (healthy or diseased), and species (human or mouse). In the mouse retina case, MarsGT identified not only six major cell populations but also a rare sub-cell population of Müller glia cells, a discovery unachievable by alternative computational tools and unreported in the original study. In the human small lymphocytic lymphoma case, MarsGT pinpointed a rare B cell lymphoma population, with unique transcription factors and binding enhancer changes indicating potential regulatory mechanisms, functional differences, and a possible precursor state for B cell lymphoma. This finding could lead to early detection or prevention strategies for B cell lymphoma progression. In the melanoma case, MarsGT identified two CD8+ Mucosal-associated invariant T (MAIT)-like rare subpopulations and revealed that high IFN-I response hinders these MAIT-like cell responses by upregulating IL10 and inducing IL15 and IL12 from DC in patients who responded to immune checkpoint blockade. These examples underscore the prowess of MarsGT in uncovering new biological insights, as well as generating new biomarkers for guiding immunotherapy.

While MarsGT shows impressive performance in identifying rare cells and uncovering biological insights, there is room for improvement. Statistical significance is crucial for rare cell identification to ensure that

detected rare cell populations genuinely exist, rather than being random false positives. In this study, we utilized scPOWER to maintain high-confidence rare cell populations. It is necessary to develop a new method for significance testing. More complex scenarios, such as senescent cells that exhibit high heterogeneity even within the same cell type, also need to be considered. For multi-sample datasets, batch correction is necessary prior to the MarsGT application. Thus, developing an algorithm that can perform batch correction during training could enhance the utility of MarsGT. Moreover, the model's dependency on GPU computations might challenge reproducibility. While our benchmark tests demonstrate negligible variance across multiple runs, the small number of rare cell populations identified suggests that the focus should be on significant findings for analysis. Finally, although we designed a regularization term to balance the signal between rare and major cell populations, this approach did sacrifice some performance in identifying major cell populations to ensure accuracy in rare cell identification (Supplementary Fig. 22). More intricate designs should be considered in future work. In conclusion, MarsGT represents a tool for the identification of rare cell populations and the elucidation of microenvironmental and immunotherapeutic mechanisms. It sets a promising trajectory for precision medicine by enabling the discovery of disease-associated rare cell populations and uncovering intrinsic regulatory mechanisms that could inform immunotherapy strategies.

## Methods

### Data preprocessing

MarsGT initiates by inputting the raw count matrices derived from matched scRNA-seq $\mathbf{X^R} = \{x_{ik}^R | i = 1, 2, \ldots, M_1; k = 1, 2, \ldots, N\}$ and scATAC-seq $\mathbf{X^A} = \{x_{jk}^A | j = 1, 2, \ldots, M_2; k = 1, 2, \ldots, N\}$. For the scRNA-seq data matrix, we organize it such that rows represent genes, whereas cells constitute the columns. Conversely, the scATAC-seq data matrix is structured with regulatory regions (peaks) as rows and cells as columns. Any row or column in each data matrix containing less than 0.1% non-zero values is excluded from further analysis. Quality control measures for the data are conducted utilizing Seurat v3[54], encompassing criteria like total read counts and mitochondrial gene ratios.

We then construct the regulatory score matrix $\mathbf{X^{RA}} = \{x_{ij}^{RA} | i = 1, 2, \ldots, M_1; j = 1, 2, \ldots, M_2\}$ based on MAESTRO[55]. In this matrix, $x_{ij}^{RA}$ signifies the regulatory potential of peak $j$ relative to gene $i$. This potential is determined in accordance with the genomic distance between peak $j$ and gene $i$.

$$x_{ij}^{RA} = \begin{cases} 0, d_{ij} > 150kb \text{ or peak } j \text{ located in any nearby genes} \\ \frac{1}{Length(exon)}, \text{peak } j \text{ located at the exon regions of the gene } i \\ 2^{-\frac{d_{ij}}{d_0}}, \text{else} \end{cases}$$

$$(1)$$

The distance between the center of peak $j$ and the transcription start site of gene $i$ is denoted as $d_{ij}$. The half-decay of the distance, $d_0$, is set to be 10 kb. $Length(exon)$ is the length of the exon where the peak $j$ located in. As indicated by formula (1), the regulatory potential score $x_{ij}^{RA}$ of peak $j$ relative to gene $i$ is typically calculated by $2^{-\frac{d_{ij}}{d_0}}$; For

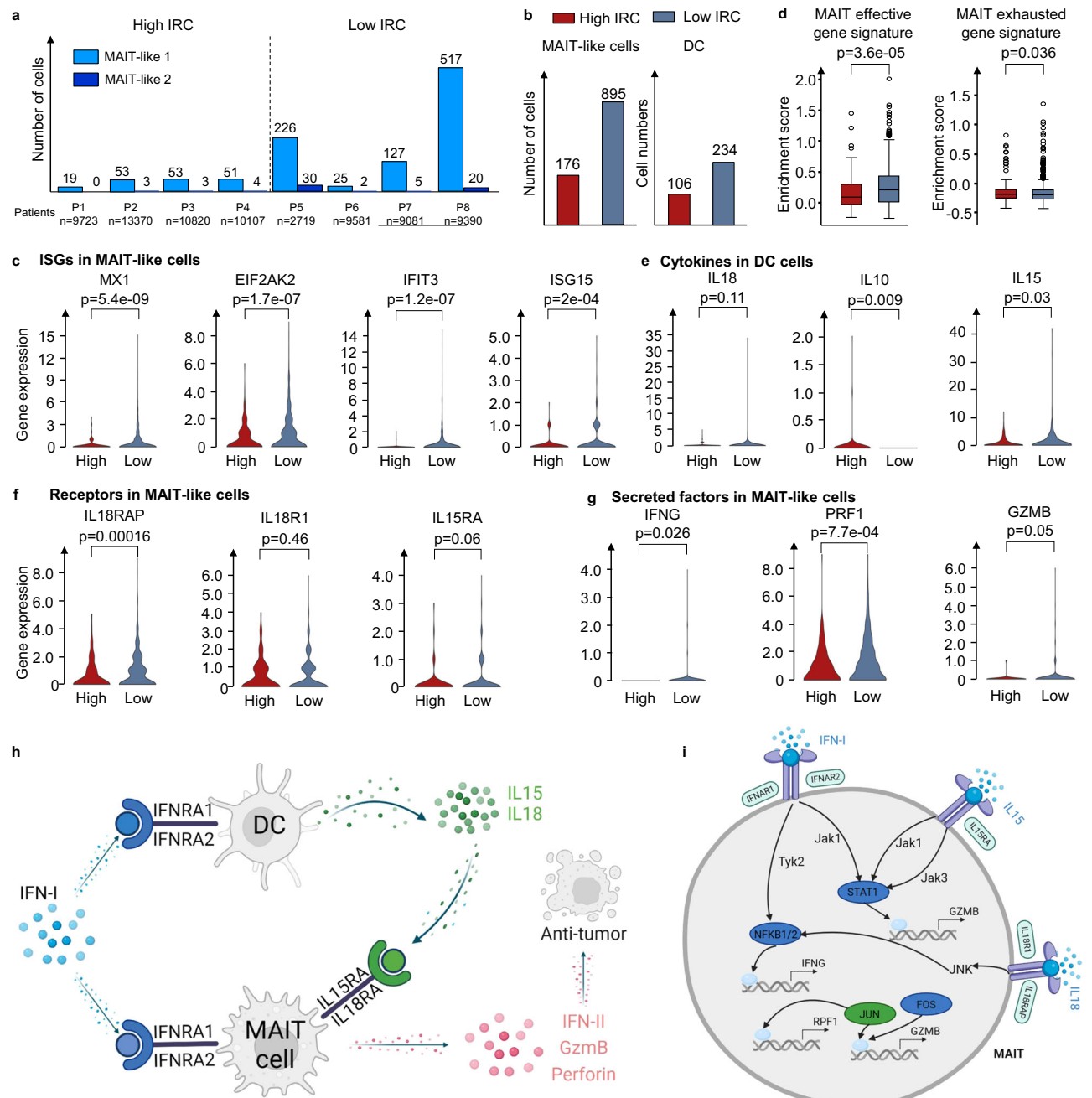

**Fig. 6 | MarsGT reveals the good prognosis with high IFN-I response impairs MAIT cell responses by increasing IL10 and inducing IL15 and IL12 from DC.**
**a** The number of cells in MAIT-like cell populations in different melanoma patients.
**b** The number of cells in MAIT cells and DC cells in high IRC and low IRC.
**c** Expression of ISGs in MAIT cells across different IRC. **d** Enrichment score of MAIT effective gene signature and exhausted gene signature across varying IRC. Each box showcases the minimum, first quartile, median, third quartile, and maximum gene signature enrichment scores in different IRC in MAIT cells (High IRC: $n = 176$, Low IRC: $n = 895$). The $p$-value is calculated by the two-sided Mann-Whitney U test.

**e** Expression of critical cytokine genes in DC cells across varying IRC. **f** Expression of IFN-I receptor coding genes in MAIT cells across different IRC. **g** Expression of secreted factors coding genes in MAIT cells across different IRC. **h** The mechanism of MAIT-like cells with different IRC samples. **i** The regulatory mechanism of IFN-II, GzmB, and Perforin in high IRC and low IRC. Green means the common regulatory relations between high IRC and low IRC. Dark blue means the regulatory relations are unique in low IRC. Figure created with BioRender.com. Source data are provided as a Source Data file.

peaks with $d_{ij} > 150kb$, we conveniently assign the regulatory potential score of 0, considering that it will be less than 0.0005. For peaks located within the exon region, $x_{ij}^{RA}$ is computed as $\frac{1}{Length(exon)}$.

**Multiple datasets integration**
In our specific case analysis, we handled matched scRNA-seq and scATAC-seq across multiple samples. For batch effect correction in multiple scRNA-seq datasets, we employed Harmony[56], resulting in an integrated matrix. For multiple scATAC-seq datasets, we adopted a binning approach with a length of 5000 base pairs to amalgamate different samples. The counts of peaks falling within the same bin were aggregated. Subsequently, multiple scATAC-seq datasets were integrated into a single matrix, where rows represented bins and columns denoted cells.

## MarsGT model's construction

**Heterogeneous graph construction.** To model and represent the scMulti-omics data, a heterogeneous graph is constructed, comprising nodes of cells, genes, and peaks. The intuitions of creating such a heterogeneous graph are: (1) Gene and peak entities do not exist in isolation. They interact with each other in intricate ways in cells. The heterogeneous graph captures these interactions and dependencies into a unified framework. (2) A heterogeneous graph enables the identification of joint embeddings of cells, genes, and peaks in a holistic manner. Such joint embeddings will benefit the harmonization of the two data sources, revealing cross-modal relationships that might be missed when analyzing each data type in isolation. (3) A heterogeneous graph also allows the message to pass across different cells, genes, and peaks. The local and global message passing and transferring in the later graph transformer model can minimize the effect of missing values or dropout issue in single-cell data.

In our case, we integrate matrices $\mathbf{X^R}$ and $\mathbf{X^A}$ by constructing a gene-cell-peak heterogeneous graph $\mathbf{G}$, consisting of three node types and four edge types to ensure each type of element (nodes and edges) maintains a unique distribution and furnishes a natural representation framework. We define the heterogeneous graph as $\mathbf{G}=(\mathbf{V},\mathbf{E},\mathbf{F})$ with node set $\mathbf{V}=\mathbf{V^C}\cup\mathbf{V^E}\cup\mathbf{V^G}$, where $\mathbf{V^C}=\{v_k^C|k=1,2,\ldots,N\}$ denotes all cells, $\mathbf{V^E}=\{v_j^E|j=1,2,\ldots,M_2\}$ denotes all peaks, $\mathbf{V^G}=\{v_i^G|i=1,2,\ldots,M_1\}$ denotes all genes. The edge set $\mathbf{E}$ is constituted as $\{(v_i^G,v_k^C),(v_k^C,v_i^G),(v_j^E,v_k^C),(v_k^C,v_j^E)|i=1,2,\ldots,M_1 j=1,2,\ldots,M_2,k=1,2,\ldots,N\}$, with edge weight $w$ defined as follows. To eliminate information redundancy between node initial embeddings and the edge weights, we utilize unweighted edges when constructing the heterogeneous graph. For $X_{ik}^R>0, w(v_i^G,v_k^C)=w(v_k^C,v_i^G)=1$, otherwise, $w(v_i^G,v_k^C)=w(v_k^C,v_i^G)=0$. For $X_{jk}^A>0$, $w(v_j^E,v_k^C)=w(v_k^C,v_j^E)=1$, otherwise, $w(v_j^E,v_k^C)=w(v_k^C,v_j^E)=0$. Lastly, we establish the initial feature vectors $\mathbf{F}$ for nodes in $G$ as follows:

$$F_k^C=X_{\cdot,k}^R, k=1,2,\ldots,N;$$

$$F_j^E=\left(X_{j,\cdot}^A\right)^T, j=1,2,\ldots,M_2;$$

$$F_i^G=\left(X_{i,\cdot}^R\right)^T, i=1,2,\ldots,M_1;$$

where $X_{i,\cdot}$ and $X_{\cdot,k}$ represent the $i_{th}$ row vector and the $k_{th}$ column vector of $\mathbf{X}$, respectively.

**Sub-sampling of a heterogeneous graph.** To enhance the efficiency of MarsGT when dealing with a large heterogeneous graph, it is necessary to select subgraphs prior to model training. We assume that a gene ubiquitously expressed is unlikely to hold as much significance for rare cell identification compared to a gene that is expressed only in a particular subpopulation. To discern rare cells, we devised a probability-based sub-sampling method. It is imperative to identify genes or peaks that are highly expressed in a target cell but exhibit low or no expression in other cells. There are two steps for the probability-based sub-sampling method. The first step is to filter out lowly expressed genes, which should not be regarded as rare-related features. For cell $k_0$, the genes $v_{ik_0}^G$ with $x_{ik_0}^G>a$ are reserved. $a$ is a threshold set to the first quartile of the expression value of all genes in the given cell.

The second step is to select genes and peaks based on probabilities calculated by the following formula:

$$Prob\left(v_{ik_0}^G\right)=\frac{Prop\left(v_{ik_0}^G\right)}{\sum\limits_{\{i|x_{ik_0}^G>a\}}Prop\left(v_{ik_0}^G\right)} \tag{2}$$

where

$$Prop\left(v_{ik_0}^G\right)=\frac{x_{ik_0}^R}{\sum\limits_k x_{ik}^R} \tag{3}$$

The greater the probability value of gene correspondence, the easier it is to be selected into the subgraph of the corresponding cell. At this time, the gene has a higher probability of displaying the rare signal of the cell. The number of reserved genes is denoted as $N_g$. Considering the expensive nature of the deep learning model, here we set $\min(N_g,20)$ number of genes are selected as default. As $\mathbf{X^A}$ trends towards binarization, peaks $v_{jk_0}^E$ within the cell according to a probability that is defined as follows:

$$Prob\left(v_{jk_0}^A\right)=\frac{Prop\left(v_{jk_0}^A\right)}{\sum\limits_j Prop\left(v_{jk_0}^A\right)} \tag{4}$$

where

$$Prop\left(v_{jk_0}^A\right)=\frac{x_{jk_0}^A}{\sum\limits_k x_{jk}^A} \tag{5}$$

The greater the probability value of peak correspondence, the easier it is to be selected into the subgraph of the corresponding cell. At this time, the peak has a higher probability of displaying the rare signal of the cell. The number of reserved peaks is denoted as $N_p$. Considering the expensive nature of the deep learning model, here we set $\min(N_p,20)$ number of peaks as default.

Each subgraph incorporates 30 cells randomly along with their selected neighbor nodes. MarsGT is trained using multiple mini-batches, each represented by a subgraph.

**MarsGT embedding update.** Let $\mathbf{H^l}$ represent the embedding of the $l^{th}$ layer ($l=1, 2,\ldots, L$). The updated embedding of $v_i^G, v_j^E, v_k^C$ on the $l^{th}$ the layer is denoted as $H^l[v_i^G]$, $H^l[v_j^E]$, and $H^l[v_k^C]$, respectively. To align the features of different types of nodes into the same dimension, we apply a linear projection function $\mathbf{W}$ on the initial feature vectors $\mathbf{F}$ and obtain the initial embeddings $\mathbf{H^0}$ with a lower dimension, which we set at 256:

$$H^0\left[v_i^G\right]=W_G\left(F_i^G\right) \tag{6}$$

$$H^0\left[v_j^E\right]=W_E\left(F_i^E\right) \tag{7}$$

$$H^0\left[v_k^C\right]=W_C\left(F_i^C\right) \tag{8}$$

Subsequently, we apply a multi-head mechanism to divide $H^0[v]$ evenly into $\mathbf{H}$ heads. Within the $l^{th}$ layer, we build three linear projection functions, query ($Q_{linear}^h$), key ($K_{linear}^h$), and value ($V_{linear}^h$), for each head ($h=1,\ldots,H$). For each node $v$ within the graph, the feature vectors obtained after these transformations are denoted as follows:

$$Q^h(v)=Q_{linear}^h\left(H^{l-1}[v]\right) \tag{9}$$

$$K^h(v)=K_{linear}^h\left(H^{l-1}[v]\right) \tag{10}$$

$$V^h(v)=V_{linear}^h\left(H^{l-1}[v]\right) \tag{11}$$

To calculate the mutual attention between node $v$ and its neighbor $N(v)$ within the $h^{th}$ head, we introduce the attention operator. This operator estimates the importance of each neighboring node $v^{ne}$ in $N(v)$ relative to $v$ using $K^h(v^{ne})W_{t(v^{ne},v)}^{ATT}Q^h(v)^T$, where $W_{t(v^{ne},v)}^{ATT}$ is a transformation matrix designed to capture edge features, while $t(.)$ denotes the edge type. $(.)^T$ signifies the transposal function. The attention coefficient within the head $h_0$ is then calculated as follows:

$$att(v^{ne},v,h_0) = \underset{all\ v^{ne}\in N(v)}{Softmax}\left(K^{h_0}(v^{ne})W_{t(v^{ne},v)}^{ATT}Q^{h_0}(v)^T\right) \quad (12)$$

The concatenation of attention heads yields the attention coefficients, represented as follows:

$$att(v^{ne},v) = \underset{h}{||}\left(att(v^{ne},v,h)\right) \quad (13)$$

The message from $v^{ne}$ that can be relayed to $v$ within head $h$ is given by $V^h(v^{ne})W_{t(v^{ne},v)}^{MSG}$. $W_{t(v^{ne},v)}^{MSG}$ is also a transformation matrix. Then, the results from different message heads should subsequently be concatenated:

$$mes(v^{ne},v) = \underset{h}{||}V^h(v^{ne})W_{t(v^{ne},v)}^{MSG} \quad (14)$$

To update the embedding of node $v$, the final step within the $l^{th}$ layer will sum $H^{l-1}[v]$ and $H^{l'}[v]$ with trainable weights into the node's new embedding.

$$H^{l'}[v] = \underset{\forall v^{ne}\in N(v)}{Aggregate}\ att(v^{ne},v)mes(v^{ne},v) \quad (15)$$

$$H^l[v] = \alpha ReLU\left(H^{l'}[v]\right) + (1-\alpha)H^{l-1}[v] \quad (16)$$

where $\alpha$ represents a trainable parameter, while ReLU functions as the activation function. The final embedding of $v$ is obtained by layer-wise stacking of information.

**MarsGT subgraph training.** For the sake of generality, we continue using the aforementioned notation for subgraphs. Following the calculation of embeddings, all nodes (genes, cells, and peaks) acquire new embeddings, denoted as $\{H^l[v]|v\in V^C\cup V^E\cup V^G\}$. The update embeddings of cells $\{H^l[v]|v\in V^C\}$ is denoted as **P** after normalizing by the sum of columns. Each row of **P** represents a cell, each column of **P** represents a reduced dimension set manually, and each element signifies the probability that a cell belongs to a specific cell cluster. We establish an initial number of cell clusters that align with the number of cell embeddings. A cell is assigned to the cell cluster where the corresponding dimension yields the highest value relative to all other dimensions. Thus, MarsGT does not require pre-specification of the number of cell clusters. Similarly, we construct the initial embedding of links between genes and peaks for each cell cluster by concatenating gene and peak embeddings, denoted as **Q**. The row of **Q** represents a peak-gene link, and the column of **Q** represents the cell cluster. By applying a linear layer and a ReLU layer, the dimension of **Q** is reduced to match the number of initial cell clusters. The output is the probability that a peak-gene link belongs to each cell cluster which is denoted as $\hat{O}$. The base peak-gene relations are determined based on $X^{RA}$ and adjusted according to the corresponding gene expressions and chromatin accessibility of all cells in a cell cluster, denoted as **O**.

In our model training, we devise a multi-task loss function, which consists of four critical components (Supplementary Fig. 2). Loss components (1) and (2) are designed to obtain high-quality node embeddings. Loss components (3) and (4) are the key to identifying rare and major cell populations simultaneously. Based on component (3), we introduce the peak-gene relations in the model training

process, which can provide more accurate rare signals. Component (4) is critical for our model to keep the major population signal not diminished while we focus on rare populations. The multi-task loss function is defined as follows:

$$Loss = \overset{(1)}{\overbrace{KL_{cluster}}} - \overset{(2)}{\overbrace{Cos_{loss}}} + \overset{(3)}{\overbrace{KL(\hat{O},O)}} + \overset{(4)}{\overbrace{\delta Reg_{loss}}} \quad (17)$$

where $KL_{cluster} = KL(H^l[V^G]*H^l[V^C]^T,X^R) + KL(H^l[V^E]*H^l[V^C]^T,X^R)$. $\delta$ is the weight coefficients, the default values are all set to 1.

The Cosine similarity in cluster $C_0$ is defined as:

$$Cos_{loss}(C_0) = \sum_{\forall k_a,k_b\in C_0} Cosine(H^l[V^{k_a}],H^l[V^{k_b}]) \quad (18)$$

The regular term is defined as:

$$Reg_{loss} = Smoothing_{cross(Pr,L,\varepsilon)} \quad (19)$$

where $L = \{l_j|l_j\in\{1,\dots,T\},j=1,\dots,N\}$ is the cell cluster results by Louvain with scRNA-seq, **Pr** is the predicted results of model. $T$ is the number of cell clusters by Louvain, $\varepsilon$ is a smoothing factor.

$$Smoothing_{cross(P,L,\varepsilon)} = -\sum_{j=1}^{N}\sum_{t=1}^{T}y_{jt}\log p_{jt} \quad (20)$$

$$y_{jt} = \begin{cases} 1-\varepsilon+\frac{\varepsilon}{T}\ if\left(l_j=t\right) \\ \frac{\varepsilon}{T}\ if\left(l_j\neq t\right) \end{cases} \quad (21)$$

where $p_{jt}$ represents the predicted probability of the given cell belonging to the class $t$, i.e. the corresponding element in matrix **P**.

(1) The goal of Component 1 is to maximize embedding retention of the original information of the data. In detail, $KL_{cluster} = KL(H^l[V^G]*H^l[V^C]^T,X^R) + KL(H^l[V^E]*H^l[V^C]^T,X^A)$, $H^l[V^G]$ is the embedding of genes, $H^l[V^C]$ is the embedding of cells, $H^l[V^E]$ is the embedding of peaks, $X^R$ is the expression data, and $X^A$ is the accessibility data. We expect that their inner product can greatly restore the distribution of the original expression data and accessibility data, thus preserving the original expression information. As the KL divergence diminishes, less information is lost during heterogeneous graph transformer learning.

(2) The goal of Component 2 is to ensure that the embeddings within the same cluster we identify are sufficiently similar. In detail, $Cos_{loss} = \sum_{\forall k_a,k_b\in C_0} Cosine(H^l[V^{k_a}],H^l[V^{k_b}])$. A higher similarity score indicates superior clustering efficacy and closely distance in the same cluster.

(3) The goal of Component 3 is to ensure that the union of peak-gene relations under the predicted cell cluster is similar to the peak-gene relations of bulk level, and to ensure that if a peak regulates a gene under a cell type, then the peak is accessible and the gene is expressed. We utilize another KL divergence score to evaluate discrepancies between predicted and baseline peak-gene links within each cell cluster. The baseline peak-gene associations are determined based on proximity to the nearest genes. Further, we calculate a regulatory potential score **O**, and adjust this score according to the corresponding gene expressions and chromatin accessibility across all cells within a cell cluster. Intuitively, this component can potentially reduce the false positive of cell clusters' results.

(4) The goal of Component 4 is to ensure that the feature of major cell population is not diminished, although we have specific designs for rare populations as above. In detail, $Reg_{loss} = Smoothing\ cross(Pr,L,\varepsilon)$. We calculate an entropy score

to contrast the differences between the base (Louvain clustering results, **L**) and predicted cell clustering outcomes **P**. Since the result of Louvain is not a true label, we allow for error $\varepsilon$ in calculating the cross entropy.

Our algorithm concurrently updates the cell clustering outcomes and the peak-gene links within each set of cell cluster prediction results. The model is designed to iterate until it reaches a state of convergence, at which point we obtain the cell clusters and all peak-gene links specific to each cluster. Following this, the peak-gene links within each cell cluster are obtained.

**MarsGT predicts cell clusters and eGRN in the whole graph.** Upon completing the two phases of training, a robustly trained model is generated. To ensure that every cell is mapped to its corresponding predicted cluster, and every gene and peak is associated with cell cluster-specific peak-gene regulatory information, we apply the trained model to the entire graph. By covering all cells through the chosen union of subgraphs, the trained model, when applied to the entire graph, can also have a good performance. The cell cluster predictions encompass all cells. Regarding cell cluster-specific peak-gene link, we use the final predicted cell cluster results to calculate all genes and peaks, eschewing the use of a subgraph. In order to quantify the specific degree of each peak-gene link, we determine the peak-gene score based on gene expression, peak accessibility, and the regulatory potential of peaks to genes. The peak-gene link score (PGS) is defined as:

$$PGS(i,j,CT) = \left\{ x_{i \times j,CT} = \frac{\sum_{k=1}^{|CT|} x_{ik}^{R} \times x_{ij}^{RA} \times x_{jk}^{A}}{|CT|}, \middle| i=1,2,\ldots,M_1\, j=1,2,\ldots,M_2\, k=1,2,\ldots,N \right\}$$

(22)

where $|CT|$ is the cell number in cell cluster $CT$. Then the cell cluster corresponding peak-gene links are inferred. To infer eGRN, we need to introduce the TF information. We retrieved the genome browser track file from JASPAR, which stores all known TF binding sites of each TF. A *p*-value score was provided in JASPAR. We removed TF binding sites with p-value scores more than 0.05. And then, if a TF binding site overlaps with any peak regions in the predicted peak-gene link, it will be kept, otherwise, removed. Finally, the TF-peak relations will be obtained, and the eGRNs in each cell cluster also are inferred.

## Benchmark of rare cell population identification

**Simulated single-cell multi-omics data.** We evaluated the performance of rare cell identification based on the algorithm's capability to distinguish between two known rare populations. We assessed the algorithm's efficacy using 400 simulated datasets with eight different designs from two different data types. The first data type consisted of cell line data (549 cells) characterized by low intra-class heterogeneity, available at https://trace.ncbi.nlm.nih.gov/Traces/sra/sra.cgi?study=SRP136421. This data contained five cell line types: PDX1, PDX2, HeLa.S3, K562, and HCT116. The second type was the immune population dataset (17,243 cells) with high intra-class heterogeneity, obtainable from https://www.ncbi.nlm.nih.gov/geo/query/acc.cgi?acc=GSE194122. In the cell line dataset, we selected PDX1 (176 cells) and PDX2 (167 cells) as the common cell populations, while HeLa.S3 (42 cells) and K562 (74 cells) were chosen as the rare population in the simulated data. Two types of simulated datasets were created based on the cell line data: (1) PDX1 and PDX2 (major, 290 cells), HeLa.S3 (rare, 10 cells); and (2) PDX1 and PDX2 (major, 280 cells), HeLa.S3 (rare, 10 cells), K562 (rare, 10 cells). (Supplementary Table S1). Each dataset consisted of 50 datasets, each containing 300 cells that were randomly subsampled from each cell type. Similarly, for PBMC cells, we utilized NK, CD4+ T naïve, CD8+ T (common), Erythroblast, Plasma, HSC, and

ID2-hi myeloid prog cells (rare) to establish six types of simulated datasets (Supplementary Table S1).

There were six datasets on the immune population data: (1) CD8+ T (major, 490 cells) and Plasma (rare, 10); (2) CD4+ T naïve (major, 480 cells), HSC (rare, 10 cells), and Plasma (rare, 10 cells); (3) CD8+ T (major, 490 cells), Erythroblast (rare, 10 cells), (4) CD8+ T (major, 480 cells), Erythroblast (rare, 10 cells), and HSC (rare, 10 cells); (5) CD8+ T (major, 480 cells), Erythroblast (rare, 10), and Naive CD20+ B (rare, 10); (6) CD14+Mono (major, 250 cells) and CD8+ (major, 250 cells). There were 50 datasets of 500 cells generated by randomly subsampling from each cell type.

To assess the proportion of rare cells detected by different software, datasets were created to include from 970 to 995 common cells and 30 to 5 rare cells. For instance, the smallest common dataset was comprised of 970 common cells and 30 rare cells (3%), while the largest included 995 common cells and 5 rare cells (0.5%). Six datasets of 1000 cells, with differing rare cell proportions, were generated by random subsampling from each cell type. The first five datasets tested rare cell population identification, while the sixth assessed the false-positive rate of each algorithm. All algorithms were applied to these datasets with default or suggested settings. The results were visualized using UMAP for Scanpy and colored by predicted results.

Furthermore, to ensure that the simulation datasets more closely resemble real data, we increased the number of cell types and the number of cells We simulated 150 datasets using peripheral blood mononuclear cells. Each simulation dataset contained 5,000 cells and 5-15 cell types (Sim-PBMC 7, 8, 9). The first type contains one rare cell population and four major cell populations (Sim-PBMC 7). The second type contains one rare cell population and nine major cell populations (Sim-PBMC 8). The third type contains five rare cell populations and ten major cell populations (Sim-PBMC 9). The details of each simulation dataset are shown as follows:

Sim-PBMC 7:

We have established a specific set of sampling rules to construct the simulated dataset, Sim-PBMC 7. Firstly, we definitively select the four cell types that have the highest proportions in terms of the cell numbers of each cell type. Secondly, from the two least abundant cell types (rare types), we randomly select one. Ultimately, we extract a total of 5000 cell samples from these five cell types. It is noteworthy that, except for the rare cell type, which has a fixed quantity of 50, the sampling ratios of the remaining cell types are normalized based on their proportions in the original dataset.

Sim-PBMC 8:

To verify whether our software can accurately identify rare types among numerous cell types, we have established a specific set of sampling rules to construct the simulated dataset Sim-PBMC 8. Firstly, we definitively select the nine cell types that have the highest proportions in terms of the cell numbers of each cell type. Secondly, from the two least abundant cell types (rare types), we randomly select one. Ultimately, we extract a total of 5000 cell samples from these 10 cell types. It is noteworthy that, except for the rare type of cells, which have a fixed quantity of 50, the sampling ratios of the remaining cell types are normalized based on their proportions in the original dataset.

Sim-PBMC 9:

To further increase the level of difficulty, we aim to continue verifying whether our model remains efficient when multiple rare cell types are present. We have established a specific set of sampling rules to construct the simulated dataset Sim-PBMC 9. First, we decisively choose the ten cell types that have the highest proportions in terms of the cell numbers of each cell type. Next, for those cell types that account for less than 1% of the total, we randomly select five. Ultimately, we draw a total of 5000 cell samples from these 15 cell types. It is noteworthy that the sampling ratio of each cell type is normalized based on their proportions in the original dataset.

## Simulated datasets benchmarking quantification

To optimize the default parameters of MarsGT, we conducted a grid-search test on seven simulated datasets. The parameters considered in this optimization included weighted decay (0, 0.1, 0.3), learning rate (0.001, 0.0005), and label smoothing (0, 0.1, 0.3), which are important parameters to prevent overfitting and ensure convergence speed. This resulted in a total of 18 unique parameter combinations. We randomly selected one dataset from each type of simulation as a training set. The most effective parameter combination was weighted decay = 0.1, learning rate = 0.001, and label smoothing = 0.3. This set was subsequently adopted as the default parameter combination. The remaining simulation datasets were processed using these default parameters.

## Real datasets benchmarking quantification

To assess the generalizability of our model, we applied the same parameter combinations that we used with the simulated datasets to three benchmark (PBMC-bench-1, 2, 3) real datasets. We subsequently employed the default parameters for an independent test dataset (PBMC-test).

## The ability of rare cell identification

Typically, a cell type is classified as rare if it constitutes less than 3% of the total cell count. However, truly rare cell types often represent much less than this threshold. To evaluate the capacity of various tools to identify these rare cells, we designated cell types representing 0.5%, 1%, 2%, and 3% of the total cell count as rare in the independent test dataset. We then calculated the F1 score, precision, and recall score for each category.

## Statistics & reproducibility

In our case analysis, we utilized scPower, a tool that assists in determining the power needed to detect a sufficient number of cells from a specific cell type in each individual. Using this tool, we calculated the minimum number of cells per individual required to reach a predetermined power threshold. This calculation was based on the negative binomial distribution.

No data were excluded from the analyses. The experiments were not randomized. The Investigators were not blinded to allocation during experiments and outcome assessment.

To evaluate the stability and reproducibility of MarsGT, we executed the algorithm 20 times on the independent test dataset, utilizing the default parameters. We then computed the variance of several key metrics, including the F1 score, precision, recall score, Normalized Mutual Information (NMI), purity, and entropy.

## Baseline tools parameter set

To evaluate the performance of MarsGT relative to other tools for identifying rare cells, we conducted a comparative analysis between MarsGT and other established methods.

(i) FIRE[4] (v 1.0.1, https://github.com/princethewinner/FIRE, data pre-processing, and feature extraction uses the function "ranger_preprocess").

(ii) GapClust[13] (v 0.1.0, https://github.com/fabotao/GapClust, genes that were expressed in less than three cells were excluded, and cells expressing <200 genes were also excluded, the normalization procedure was accomplished using the scran (R package).

(iii) CellSIUS[26] (v 1.0.0, https://github.com/Novartis/CellSIUS, cells were filtered based on the total number of detected genes, total UMI counts, and the percentage of total UMI counts attributed to mitochondrial genes, genes have to present with at least 3 UMIs in at least one cell. After this initial QC, the remaining outlier cells were identified and removed using the plotPCA function from the scatter (R package with detect_outliers set to TRUE). Data were normalized using scran (R package), including a first clustering step as implemented in the "quickCluster" function).

(iv) RaceID[2] (v 0.2.3, https://github.com/dgrun/RaceID3_StemID2_package).

(v) GiniClust[15] (v 3.0, https://github.com/rdong08/GiniClust3, the "neighbors" parameter of the function "clusterGini" is set to 10 (the recommended value is from 5 to 15), other parameters at the default values).

(vi) SCMER[1] (v 0.1.0a3, https://github.com/KChen-lab/SCMER, data pre-processing is carried out using Scanpy (Python package)).

For each benchmarking tools, grid tests were also applied to a combination of parameters (Supplementary Data 5).

## Evaluation index

To assess the precision of various rare cell identification algorithms, we employed metrics quantifying the purity of the clustering output. We evaluated two categories of algorithms: the first encompassing clustering methods capable of differentiating all cell populations, such as SCMER[1], GiniClust[15], and RaceID[2], and the second consisting of classification methods that can distinguish only between rare cell and major cell populations. For evaluating clustering methods, we utilized purity, entropy, and Normalized Mutual Information (NMI). For assessing classification methods, we applied recall, precision, and F1-score metrics.

**Purity** is based on the frequency of the most abundant class in the predicted clusters. Let $\mathbf{S} = \{s_1, s_2, \ldots, s_S\}$ be the set of predicted clusters, and $\mathbf{T} = \{t_1, t_2, \ldots, t_T\}$ be the set of true labels. The purity index is defined as:

$$purity(\mathbf{S},\mathbf{T}) = \frac{\sum_s \max_t |s_s \cap t_t|}{N} \tag{23}$$

where $s_s$ $(s = 1, \ldots, S)$ is the set of cells in the predicted clusters. $t_t$ $(t = 1, \ldots, T)$ is the set of cells in the true labels. $N$ is the number of cells. The value of purity ranges from 0 to 1, where 1 provides the best clustering effect.

**Entropy** uses Shannon entropy to evaluate cluster accuracy by measuring the expected amount of information from the clusters. Let $\mathbf{S} = \{s_1, s_2, \ldots, s_S\}$ be the set of predicted clusters, and $\mathbf{T} = \{t_1, t_2, \ldots, t_T\}$ be the set of true labels. The entropy of each predicted cluster $s$ is defined as:

$$H(s_s) = -\sum_t \frac{|s_{st}|}{|s_s|} \log \frac{|s_{st}|}{|s_s|} \tag{24}$$

$$s_{st} = s_s \cap t_t \tag{25}$$

Then, the entropy for all clusters is defined as:

$$entropy(\mathbf{S},\mathbf{T}) = \sum_s \frac{|s_s|}{N} H(s_s) \tag{26}$$

Lower entropy means higher clustering accuracy.

**NMI** measures the normalized dependency of the true labels on the predicted cluster. Mutual information is defined as:

$$I(\mathbf{S},\mathbf{T}) = \sum_s \sum_t \frac{|s_{st}|}{N} \log \frac{N|s_{st}|}{|s_s||t_t|} \tag{27}$$

To compare mutual information across different clusters, $I(\mathbf{S},\mathbf{T})$ is normalized to the [0,1], which is bounded by min [H(**S**),H(**T**)], where

$$H(\mathbf{S}) = -\sum_s \frac{|s_s|}{N} \log \frac{|s_s|}{N} \tag{28}$$

$$H(\mathbf{T}) = -\sum_t \frac{|t_t|}{N} \log \frac{|t_t|}{N} \tag{29}$$

Then, NMI is defined as:

$$NMI(\mathbf{S},\mathbf{T}) = \frac{I(\mathbf{S},\mathbf{T})}{\min[H(\mathbf{S}),H(\mathbf{T})]} \tag{30}$$

Higher NMI means higher clustering accuracy.

**Precision** represents the ability of the model to correctly predict rare cells among all rare cell predictions.

$$Precision = \frac{TP}{TP+FP} \tag{31}$$

**Recall** represents the model's ability to correctly predict rare cells from actual rare cells.

$$Recall = \frac{TP}{TP+FN} \tag{32}$$

**F1-score** can be interpreted as a weighted average of precision and recall. F1-score ranges from 0, poor classification, to 1, perfect classification:

$$F1\,score = \frac{TP}{TP+0.5*(TP+FN)} \tag{33}$$

$TP$ means the number of cells predicted to be rare in real rare cells. $FP$ means the number of cells predicted to be rare in real common cells. $FN$ means the number of cells predicted to be common cells in real rare cells

## Reporting summary

Further information on research design is available in the Nature Portfolio Reporting Summary linked to this article.

## Data availability

All relevant data supporting the key findings of this study are available within the article and its Supplementary Information files. All data used in this study are from public domain. The raw data are downloaded from the GEO database with the accession numbers for: human PBMC paired scRNA-seq and scATAC-seq data "GSE194122", mouse retina paired scRNA-seq and scATAC-seq data "GSE201402" and human melanoma PBMC paired scRNA-seq and scATAC-seq data "GSE199994". The scRNA-seq and scATAC-seq cancer cell line data was downloaded from NCBI with an accession code of "CNP0000213 [https://trace.ncbi.nlm.nih.gov/Traces/index.html?study=SRP136421]". The following paired scRNA-seq and scATAC-seq dataset was obtained from the 10X Genomics website: "lymph node data [https://www.10xgenomics.com/resources/datasets/fresh-frozen-lymph-node-with-b-cell-lymphoma-14-k-sorted-nuclei-1-standard-2-0-0]". All datasets are publicly available without restrictions. Details of data information can be found in Supplementary Data 1. Source data are provided with this paper.

## Code availability

MarsGT is a user-friendly, efficient package developed in Python, leveraging the capabilities of PyTorch. The source code and vignettes of MarsGT are freely available at https://github.com/mtduan/marsgt. The source code is also available on Zenodo[57] with link https://doi.org/10.5281/zenodo.8406470.

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

## Acknowledgements

This work was supported by National Key R&D Program of China (2020YFA0712400), National Nature Science Foundation of China (NSFC, 62272270 and 11931008), and Shandong University multi-disciplinary research and innovation team of young scholars (2020QNQT017). This work was supported by the Pelotonia Institute of Immuno-Oncology (PIIO). The content is solely the responsibility of the authors and does not necessarily represent the official views of the PIIO. In addition, we thank Dr. Jordan Krull from the Ohio State University for his valued suggestions in biological insights.

## Author contributions

Q.M. and B.L. conceived the basic idea. X.W. designed the algorithm, conducted the case study, and wrote the manuscript. M.D. carried out benchmark experiments and wrapped the code. A.M. and M.D. participated in the case analysis. J.L. collected the data and pre-processed the data. A.M. led the figure design. Z.L. and G.X. provided immunological knowledge. D.X. provided deep learning knowledge. All authors participated in the interpretation and writing of the manuscript.

## Competing interests

The authors declare no competing interests.
