## [Peer Review File · Nature Communications]

MarsGT: Multi-omics analysis for rare population inference using single-cell graph transformerReviewer #1 (Remarks to the Author):

Dr Xiaoying Wang et al. proposed MarsGT to infer the rare population using a probability-based heterogeneous graph transformer. To enhance the performance of rare population detection, MarsGT incorporates the advance of graph neural networks and gene regulatory networks. MarsGT was fully tested on simulation data and four human peripheral blood mononuclear cell datasets. Results show that MarsGT outperforms the existing tools in simulation data and real cases. Further, MarsGT provides approaches to not only identifying rare cell populations but also developing strategies for early clinical detection. The authors present an elegant deep-learning model with good implementation. While we welcome MarsGT to the palette of methods, I have the following questions and concerns:

1. In the overview of the MarsGT framework, the authors provide a recapitulative description of their algorithm. As the key function of MarsGT is to detect rare populations, it will be helpful to include the critical for separating rare populations from major populations.
2. In Fig 3b, rare populations are colored red based on scPower analysis. Why clusters on the bottom were not identified as rare populations?
3. Authors make a strong case for how MarsGT can identify the rare cell populations in lymph node tumors. If authors could add the scATAC-seq tracks in Fig 4 g, i, and h, it would provide more information. The height and shape of scATAC-seq may vary among cells.
4. In Fig S7, it is very interesting that only BC3 and BC6 receive the ncWNT signaling from RBC. Is there any potential explanation for this observation?

Minor:

1. In Fig. 3e, the authors used mean expression to visualize the pathway activation. This approach may be biased to high expression genes in each pathway.
2. In Fig. 4a, the figure legend has redundant characters of cluster number.
3. In supplementary Fig. 13, is there a specific reason that the color of clusters is not consistent with the rank of the clusters?
4. The writing could be improved. For example, the grammar of "MarsGT identifies rare B lymphoma-state-1 that provides the potential for preventing B-lymphoma progression" is not correct.

Reviewer #2 (Remarks to the Author):

The paper describes MarsGT an approach for the identification of rare cell types from integrated scRNA and scATAC data sets.

MarsGT constructs a heterogeneous graph from the processes input data. While a description of the graph construction is given in the Methods section, I found it difficult to fully follow the reasoning behind the concept of the gene-cell-peak graph construction and the sub-sampling then applied. I had no intuition about how the selection of 20 highly expressed genes or peaks per cell etc helps in subgraph construction (Methods sub-section "Sub-sampling of a heterogeneous graph").

As a consequence it was then difficult to fully understand what the final training objective (loss function) was. Supplementary Figure 1 has no detailed caption or accompanying text.

I did not understand how MarsGT specifically identifies rare cell types (and what the criteria were for rare cells) or if the method is generally doing unsupervised clustering and the by-product is the increased probability of identifying rare cell types.

The authors present a substantial volume of real data analysis in the latter stages of the article but I was left unsure as to how the results of their analysis would be different using another clustering algorithm?

This personal uncertainty stems from the lack of clarity in the simulation experiments. The construction of the simulated data sets could be more plainly described in the Methods or in an extended Supplementary section.

In developing the simulated data, how similar are these to "real data"? Table S1 suggests that the simulated data sets consisted of only a few hundred cells and 2-3 cell types. In all the real data sets later on, the data contains many cell types. The key challenge for any rare cell detection method is how well does it do in identifying rare types (however this is defined) and what is the trade-off in terms of potentially breaking up (or merging) other non-rare classes. It is not clear to me that the simulation set up or reported results explain the performance of MarsGT in these aspects.

It is difficult to translate the metrics of NMI, Purity and Entropy into real-world performance. A statement such as "MarsGT was 5.66% higher than the NMI score of the existing best-performing tool on the independent test dataset" is difficult to decipher. What does 5.66% translate to in practice? The results could be better demonstrated with specific graphical illustrations of simulated data experiments. In addition, it could be more useful to have something simple like the adjusted Rand index against the original labels.

When attention turns to more real-world data settings, e.g. in Figure 2D, it is difficult to understand the overall clustering performance because every method groups the cells in different ways. MarsGT appears more sensitive to the rare populations but the UMAP shows it does various things to the larger cell populations (merges cell types, or splits cell types).

What is the trade-off? How is this controlled in MarsGT?

Further, how were the ground truth labels derived? Is there uncertainty in these labels?

"Moreover, 12 rare cellular populations were distinguished, eight of which boast a 95% confidence level as highlighted by scPower"

What does this statement mean? What does the 95% confidence level from scPower refer to?

Overall, the paper presents potentially interesting methodology but from my personal perspective, the methodology description needs substantial revision to better explain and justify the design choices and most importantly provide the average reader with a more intuitive understanding of how it works. The experiments section is substantial and great efforts have been placed here but the simulated (controlled) experiment setting is the only place where the authors have full control. This section needs to be refined both in terms of description but also in the design of the experiments to match them with real world data.

Department of Biomedical Informatics

College of Medicine

3049 Pelotonia Research Center

2255 Kenny Road

Columbus, OH, 43210

Reviewer #1 (Remarks to the Author)

Dr Xiaoying Wang et al. proposed MarsGT to infer the rare population using a probability-based heterogeneous graph transformer. To enhance the performance of rare population detection, MarsGT incorporates the advance of graph neural networks and gene regulatory networks. MarsGT was fully tested on simulation data and four human peripheral blood mononuclear cell datasets. Results show that MarsGT outperforms the existing tools in simulation data and real cases. Further, MarsGT provides approaches to not only identifying rare cell populations but also developing strategies for early clinical detection. The authors present an elegant deep-learning model with good implementation. While we welcome MarsGT to the palette of methods, I have the following questions and concerns:

1. In the overview of the MarsGT framework, the authors provide a recapitulative description of their algorithm. As the key function of MarsGT is to detect rare populations, it will be helpful to include the critical for separating rare populations from major populations.

Response: Thank you for your constructive suggestion. Here, we clarify the major design for separating rare populations from major populations: a specific probability-based sub-sampling method in our heterogeneous graph representation framework. We posit that a gene ubiquitously expressed is less likely to be pivotal for identifying rare cells compared to a gene that is expressed only within a specific subpopulation. To discern rare cells, it is imperative to identify genes or peaks that are highly expressed in a target cell but exhibit low or no expression in other cells. Such genes or peaks have a higher probability of being sampled to the key features of rare cells in our multi-head attention graph transformer. The method description can be found on Page 22-23, Line 664-694 of the main manuscript. We also emphasized this aspect in the overview of the MarsGT framework section on Page 3, Lines 113-123 of the main manuscript.

2. In Fig 3b, rare populations are colored red based on scPower analysis. Why clusters on the bottom were not identified as rare populations?

Response: We are sorry for any ambiguity in our initial description and would like to provide more clarifications here. The clusters at the bottom are indeed predicted as rare cell populations by MarsGT, while they were not further analyzed in the case studies. The exclusion is carried out based on scPower, which is a tool to assist in determining the power needed to detect a sufficient number of cells from a particular cell type in each individual. Usually, a detected rare population has a very limited numerical count of cells, which is a potential risk of false-positive results and could consequently misguide the

analysis. To circumvent this risk, we further utilize scPower to calculate the minimum number of cells per individual required to attain a predetermined power threshold based on the negative binomial distribution for this calculation. Specifically, we have concentrated on rare cell populations with 95% confidence and have opted not to analyze those rare cell populations that did not achieve statistical significance in our case studies. According to your comment, we have revised Fig. 3b (left) to a new version (right), which we believe more accurately represents our data and findings.

Fig. 3b-old. The number of cells in each cell cluster. The bar with the red color represents the identified rare cell populations with a 95% confidence level.

Fig. 3b-new. The number of cells in each cell cluster. The bar with the red color represents the identified rare cell populations by MarsGT. The rare populations with a 95% confidence level are annotated on the right sidebar.

3. Authors make a strong case for how MarsGT can identify the rare cell populations in lymph node tumors. If authors could add the scATAC-seq tracks in Fig 4 g, i, and h, it would provide more information. The height and shape of scATAC-seq may vary among cells.

Response: We agree with the reviewer that including the scATAC-seq tracks will offer additional insights in Fig. 4 g, i, and h. Thus, we have added the corresponding Coverage Plot of BCL2 to Fig. 4j (see below), and the other two Coverage Plots are showcased in Supplementary Fig. 16.

Fig. 4j. The Coverage plot for gene BCL2. The Coverage Plot encompasses the tracks of scATAC-seq (upper), peak links (lower), and gene expression (right).

4. In Fig S7, it is very interesting that only BC3 and BC6 receive the ncWNT signaling from RBC. Is there any potential explanation for this observation?

Response: Thanks for this insightful question. We have identified a non-canonical Wnt (ncWnt) signaling pathway that originates from Red Blood Cells (RBC) and targets both BC3 and BC6 cell types. Prior research has substantiated that ncWnt signaling is produced by RBC [1]. Additionally, our findings are supported by recent research, which verifies that ncWnt signaling, originating from RBC, is received by BC [2]. To our best knowledge, no existing literature categorizes this signaling pathway into subtypes of BC. This gap underscores the potential significance of our work in identifying rare cell types. By increasing the differentiation of signaling pathways, our approach may offer an enhanced understanding of the interactions between cell types. We anticipate that further research may unearth new insights, building upon our method.

[1] Sarin, S. et al. Role for Wnt Signaling in Retinal Neuropil Development: Analysis via RNA-Seq and In Vivo Somatic CRISPR Mutagenesis. *Neuron* 98, 109-126 e108, doi:10.1016/j.neuron.2018.03.004 (2018).

[2] Shah, Ruchi, et al. "Non-canonical Wnt signaling in the eye." *Progress in Retinal and Eye Research* (2022): 101149.

Minor:

1. In Fig. 3e, the authors used mean expression to visualize the pathway activation. This approach may be biased to high expression genes in each pathway.

Response: We agree with the reviewer that utilizing mean expression for visualizing pathway activation may bias the results toward genes with high expression. Therefore, we have opted to utilize gene signature enrichment score in this revision [1], which is a rank-based method to avoid biased results with high expression genes, instead of mean expression. The updated Fig. 3e is as follows.

Fig. 3e. The enrichment score across each pathway is calculated using differentially expressed genes (DEGs) across various BC subpopulations. Pathways within the red box have been validated in the literature.

Based on the new results, we focus on the analysis of four pathways, noted by the red box. Neuron migration was moderately enriched in RBC, which aligns with its translocation from the bipolar to the amacrine cell layer. Categories such as axonogenesis and the glutamate receptor signaling pathway revealed modest differences among BC clusters. Interestingly, extracellular ligand-gated ion channel activity was predominantly enriched in OFF types, reflecting their employment of ionotropic glutamate receptors [2]. We hope that these revisions and clarifications adequately address the concern raised.

[1] Andreatta, M. & Carmona, S. J. UCell: Robust and scalable single-cell gene signature scoring. *Comput Struct Biotechnol J* 19, 3796-3798, doi:10.1016/j.csbj.2021.06.043 (2021).

[2] Shekhar, K. et al. Comprehensive Classification of Retinal Bipolar Neurons by Single-Cell Transcriptomics. *Cell* 166, 1308-1323 e1330, doi:10.1016/j.cell.2016.07.054 (2016).

2. In Fig. 4a, the figure legend has redundant characters of cluster number.

Response: Thank you for your attentive observation. We have amended Fig. 4a (see below) by removing the redundant characters, ensuring a clearer presentation.

Fig. 4a. UMAP visualizes cell clusters predicted by MarsGT, annotated based on the marker genes.

3. In supplementary Fig. 13, is there a specific reason that the color of clusters is not consistent with the rank of the clusters?

Response: We have adjusted the rank convention to have consistent colors, as can be viewed in Supplementary Fig. 15 (see below).

Supplementary Fig. 15. The cell cluster results on the lymphoma dataset by Seurat. The circle means B cells, which are annotated by the curated marker genes.

4. The writing could be improved. For example, the grammar of “MarsGT identifies rare B lymphoma-state-1 that provides the potential for preventing B-lymphoma progression” is not correct.

Response: Thank you for your comment. This sentence has been changed into “MarsGT identifies a rare state, B lymphoma-state-1, which offers the potential in preventing B-lymphoma progression”. We have also checked and modified the grammar throughout the entire manuscript.

Reviewer #2 (Remarks to the Author)

The paper describes MarsGT an approach for the identification of rare cell types from integrated scRNA and scATAC data sets.

1 MarsGT constructs a heterogeneous graph from the processes input data. While a description of the graph construction is given in the Methods section, I found it difficult to fully follow the reasoning behind the concept of the gene-cell-peak graph construction and the sub-sampling then applied.

Response: Thank you for your comments. We expanded on the rationale behind two pivotal steps of our methods as below, and the corresponding content has been updated in the Methods section of the main manuscript (“Heterogeneous graph construction” on Page 21, Line 631-642 of the main manuscript and “Sub-sampling of a heterogeneous graph” on Page 22, Line 663-697 of the main manuscript).

(1) **Heterogeneous Graph Construction:** To model and represent the scMulti-omics data, a heterogeneous graph is constructed, comprising nodes of cells, genes, and peaks. The intuitions of creating such a heterogeneous graph are: (1) Gene and peak entities do not exist in isolation. They interact with each other in intricate ways in cells. The heterogeneous graph is designed to capture these interactions and dependencies in a unified framework. (2) A heterogeneous graph enables the identification of joint embeddings of cells, genes, and peaks in a holistic manner. Such joint embeddings will benefit the harmonization of the two data sources, revealing cross-modal relationships that might be missed when analyzing each data type individually. (3) A heterogeneous graph also allows the message passing across different cells, genes, and peaks. The local and global message passing and transferring in the later graph transformer model can minimize the effect of missing values or dropout issues in single-cell data.

(2) **Graph sub-sampling:** To improve the efficiency and capability of MarsGT on a giant heterogeneous graph, we selected subgraphs and training on multiple mini batches. We assume that a gene ubiquitously expressed is unlikely to hold as much significance for rare cell identification compared to a gene that is expressed only in a particular subpopulation. To discern rare cells, we devised a probability-based sub-sampling method. It is imperative to identify genes or peaks that are highly expressed in a target cell but exhibit low or no expression in other cells. We defined such genes and peaks as rare-related genes and peaks. There are two steps for the probability-based sub-sampling method. The first step is to filter out lowly expressed genes, which should not be regarded as rare-related features (Supplementary Fig. 1B ②). This step is designed to reduce false positives and accelerate computation speed. The second step is to select genes and peaks based on probabilities. The genes are selected by the following formulas:

$$Prob\left(v_{ik_0}^{G/A}\right) = \frac{Prop\left(v_{ik_0}^{G/A}\right)}{\sum_{\{i|x_{ik_0}^G > a\}} Prop\left(v_{ik_0}^{G/A}\right)} \quad (2)$$

where

$$Prop(v_{ik_0}^{G/A}) = \frac{x_{ik_0}^{R/A}}{\sum_k x_{ik}^{R/A}} \quad (3)$$

The peaks are selected by the following formula:

$$Prob(v_{jk_0}^A) = \frac{Prop(v_{jk_0}^A)}{\sum_j Prop(v_{jk_0}^A)} \quad (4)$$

where

$$Prop(v_{jk_0}^A) = \frac{x_{jk_0}^A}{\sum_k x_{jk}^A} \quad (5)$$

Such rare-related genes and peaks have a higher probability of being sampled to the key features of rare cells in our multi-head attention graph transformer (Please also check the response to Reviewer #1 Question 1). The higher the probability value of genes/peaks correspondence, the easier it is to be selected into the subgraph of the corresponding cell.

Supplementary Fig. 1B Illustration of graph sub-sampling. ① For a heterogeneous graph with seven expressed genes (rhombus) and eight accessed peaks (flags) in a given cell. ② Low expressed genes (g_5, g_6, g_7) and accessed peaks (e_5, e_6, e_7, e_8) are filtered out based on the first quartile of the expression value of all genes/peaks. The probability is calculated by the proportion of expression and accessibility. ③ Gene g_i and peak e_j are selected with $Prob(g_i)$ and $Prob(e_j)$. The higher the probability value of gene correspondence, the easier it is to be selected into the subgraph of the corresponding cell. For example, gene g_1 has a probability of 0.6 of being selected as a subgraph element for the corresponding cell. Peak e_1 has a probability of 0.5 of being selected as a subgraph element for the corresponding cell.

2 I had no intuition about how the selection of 20 highly expressed genes or peaks per cell etc helps in subgraph construction (Methods sub-section “Sub-sampling of a heterogeneous graph”).

Response: We are happy to provide future clarification here. First, we did not select the top 20 highly expressed genes or peaks per cell. Instead, our initial step was to filter out lowly expressed genes and assess peaks, which should not be regarded as rare-population-related features. The number of remaining genes is denoted as N_g , and the

number of remaining peaks is denoted as N_p . Considering the expensive computing nature of the deep learning model, here we recommend $\min(N_g/N_p, 20)$ genes/peaks. In subgraph construction, we tested different values and found that 20 can ensure robust and efficient performance for the data we used in MarsGT. Hence, we used 20 as default. This number can be adjusted by the users if there is a need in other application scenarios.

3. As a consequence it was then difficult to fully understand what the final training objective (loss function) was. Supplementary Figure 1 has no detailed caption or accompanying text.

Response: To address this concern, we explain the rationale of each of the four components in our loss function here.

$$Loss = \overbrace{KL_{cluster}}^{(1)} - \overbrace{Cos_{loss}}^{(2)} + \overbrace{KL(\hat{O}, O)}^{(3)} + \overbrace{\delta Reg_{loss}}^{(4)}$$

Loss components (1) and (2) are designed to obtain high-quality node embeddings. Loss component (3) is the key to identifying rare populations, and component (4) is the regularizer to maintain major populations while we focus on rare populations.

(1) The goal of Component 1 is to maximize embedding retention of the original information of the data. In detail, $KL_{cluster} = KL(H^l[V^G] * H^l[V^C]^T, X^R) + KL(H^l[V^E] * H^l[V^C]^T, X^A)$, $H^l[V^G]$ is the embedding of genes, $H^l[V^C]$ is the embedding of cells, $H^l[V^E]$ is the embedding of peaks, X^R is the expression data, and X^A is the accessibility data. We expect that their inner product can greatly restore the distribution of the original expression data and accessibility data, thus preserving the original expression information. As the KL divergence diminishes, less information is lost during heterogeneous graph transformer learning.

(2) The goal of Component 2 is to ensure that the embeddings within the same identified cluster are sufficiently similar. In detail, the similarity score is defined as $Cos_{loss} = \sum_{k_a, k_b \in C_0} Cosine(H^l[V^{k_a}], H^l[V^{k_b}])$. A higher similarity score indicates superior clustering efficacy and closely distance in the same cluster.

(3) The goal of Component 3 is to ensure that the union of peak-gene relations under the predicted cell cluster is similar to the peak-gene relations of bulk level, and to ensure that if a peak regulates a gene under a cell type, then the peak is accessible and the gene is expressed. Intuitively, this component can potentially reduce the false positive of cell clusters' results. We introduce the peak-gene relations in the model training process, which can provide more accurate rare signals.

(4) The goal of Component 4 is to add a regularizer to maintain the signals of major populations while we focus on rare populations. In detail, $Reg_{loss} = Smoothing_cross(P, L, \epsilon)$. We calculate an entropy score to contrast the differences between the base (Louvain clustering results, L) and predicted cell clustering outcomes P . Since the result of Louvain is not a true label, we allow for error ϵ in calculating the cross entropy. δ is the weight coefficient with a default setting of 1.

We have enriched the legend of Supplementary Fig. 1 to describe the figure better. Additionally, we have added a new figure to Supplementary Fig. 2 to elucidate the design of the loss function. (see below).

Supplementary Fig. 2. The detailed loss function explanation of MarsGT for rare cell population identification. Component 1: gene embeddings and peak embeddings interact through an inner product operation with cell embeddings, decoding the encoded information and preserving the original data (Blue color). Component 2: the cosine similarity of cell embeddings is calculated to ensure that embeddings within the same cluster are closer or more tightly knit. (Orange color). Component 3: The cell gene embeddings and peak embeddings are integrated and utilized as the input for the peak-gene relation predictor, which outputs a peak-gene link probability matrix. Subsequently, the KL divergence is calculated to ensure that the union of peak-gene relations under the predicted cell cluster is similar to the peak-gene relations of bulk level, and to ensure that if a peak regulates a gene under a cell type, then the peak is accessible and the gene is expressed (Green color). Component 4: scRNA-seq data is employed to compute the base cell cluster labels, preserving the primary cell signals (Purple color).

4. I did not understand how MarsGT specifically identifies rare cell types (and what the criteria were for rare cells) or if the method is generally doing unsupervised clustering and the by-product is the increased probability of identifying rare cell types.

Response: Thanks for your comments. MarsGT is specifically designed for rare cell type identification rather than a by-product of cell clustering. In MarsGT, to ensure genes/peaks associated with rare cell types are prioritized, we designed a specific probability-based sub-sampling method. We consider that a gene ubiquitously expressed is less likely to be pivotal for identifying rare cells compared to a gene that is expressed only within a specific subpopulation. Such genes or peaks have a higher probability of being sampled to the key features of rare cells in our multi-head attention graph

transformer. (Please also check the response to Reviewer #1 Question 1).

5. The authors present a substantial volume of real data analysis in the latter stages of the article but I was left unsure as to how the results of their analysis would be different using another clustering algorithm?

Response: We compared MarsGT with two selected tools using three datasets. The first tool is Seurat 4.0, a popular tool for cell clustering. The second tool is GiniClust, the second-best tool for rare cell identification in our benchmark section (Fig. 1a, b and Supplementary Figs.3-5). The results from these two tools in the three cases are shown as follows. If you believe the comparison results should be included in the supplementary material, we are happy to add them.

In mouse retina case, BC cells are recognized as a cell type with very many rare subtypes. In the following figure, the upper panel presents the MarsGT result, the middle panel presents the Seurat result, and the lower panel displays the GiniClust result. We can observe that Seurat only identified five BC subtypes and five unknown clusters based on the existing marker genes. For GiniClust, it only identified one BC subtype, and most cells were annotated as ROD cell types. They not only cannot identify rare cell populations, but also cannot distinguish major cell populations.

The cell cluster results of MarsGT on mouse retina case.

The cell cluster results of Seurat on mouse retina case.

The cell cluster results of GiniClust on mouse retina case.

In the diffuse small lymphocytic lymphoma case, MarsGT identified a rare state, lymphoma-state-1, which offers the potential for preventing B-lymphoma progression. Moreover, it can clearly distinguish between normal B cells and tumor B cells, as well as other major cell populations. In the following figure, the upper panel presents the MarsGT result, the middle panel presents the Seurat result, and the lower panel displays the GiniClust result. Observably, Seurat identified only two B cell types, neither of which were rare cell populations, and it also failed to distinguish between normal B and tumor B cells. Conversely, GiniClust was unable to identify a rare cell population within the B cells and also presented unknown clusters based on existing marker genes.

The cell cluster results of MarsGT on diffuse small lymphocytic lymphoma case.

The cell cluster results of Seurat on diffuse small lymphocytic lymphoma case. The circle means B cells which are annotated by the curated marker genes.

The cell cluster results of GiniClust on diffuse small lymphocytic lymphoma case. The circle means B cells which are annotated by the curated marker genes.

In the melanoma case, MarsGT identified MAIT-like rare cell populations and discovered a mechanism explaining the differences in survival following PD1-blocking immunotherapy between MAIT-like cells and DC cells. In the following figure, the upper panel presents the MarsGT result, the middle panel presents the Seurat result, and the lower panel displays the GiniClust result. It can be observed that neither of the two tools can identify rare MAIT-like cells, nor can they identify DC cells, a major cell type in the case. Additionally, Seurat has many unknown clusters based on the existing marker genes.

The cell cluster results of MarsGT on melanoma case.

The cell cluster results of Seurat on melanoma case.

The cell cluster results of GiniClust on melanoma case.

In summary, the analysis and application of real data, upon which our tool is based, cannot be achieved by other available tools (i.e., Seurat and GiniClust). If you have recommendations for additional tools, we are open to incorporating them in comparisons.

6. This personal uncertainty stems from the lack of clarity in the simulation experiments. The construction of the simulated data sets could be more plainly described in the Methods or in an extended Supplementary section.

Response: Thank you for your advice. We have revised the description to data simulation in the “Simulated single-cell multi-omics data” in Methods (Pages 26-28, Lines 840-916). The revised description includes both our original simulation methods and the new simulations as requested in Question 7 below.

7 In developing the simulated data, how similar are these to “real data”? Table S1 suggests that the simulated data sets consisted of only a few hundred cells and 2-3 cell types. In all the real data sets later on, the data contains many cell types.

Response: Thank you for your comments. We generated 150 new simulation datasets that are likely close-to-real in terms of the number of cell types and number of cells. We constructed three types of simulation datasets (Sim-PBMC 7, 8, 9), each type containing 50 simulation datasets and each simulation dataset containing 5,000 cells. The first type contains one rare cell population and four major cell populations (Sim-PBMC 7). The second type contains one rare cell population and nine major cell populations (Sim-PBMC 8). The third type contains five rare cell populations and ten major cell populations (Sim-PBMC 9). To ensure the fairness of simulation dataset construction, for each simulation dataset, the cell type and number of each cell type are randomly selected based on the ground truth cell labels and their corresponding proportions. The previous simulation results have been moved to Supplementary Figs. 3 and 5.

8 The key challenge for any rare cell detection method is how well does it do in identifying rare types (however this is defined) and what is the trade-off in terms of potentially breaking up (or merging) other non-rare classes. It is not clear to me that the simulation set up or reported results explain the performance of MarsGT in these aspects.

Response: According to our response to Reviewer #2 Question 7, the new simulation results are showcased below (also in Supplementary Fig. 4). From these figures, we conclude that MarsGT achieves the best performance in rare cell identification on all simulation datasets in terms of NMI, Purity, Entropy, F1, Precision, and Recall.

Supplementary Fig. 4. Performance comparison of the major cell and rare cell population simultaneously identification ability on 150 new simulated datasets in terms of F1 score, Precision, Recall, NMI, Purity, and Entropy.

The following figures are Sankey plots of a simulation dataset result. It is apparent that MarsGT exhibits the most exemplary performance among the other tools. Concurrently, we noticed some instances of merging or splitting within certain major cell types. To ensure accuracy in rare cell identification, we sacrificed some degree of performance in identifying major cell populations. We designed a regular term to balance the signal between rare and major cell populations (Details can be found in Reviewer #2 Question 3 loss component (4)). If a more accurate identification of rare cell types is required, we can reduce the signal of the major cell type by increasing the weight δ of the regular term during the training process, and conversely, we can enhance the weight of the term as needed (Please also check the response to Reviewer #2 Question 3 Component (4) and Question 11). Consequently, while the classification of rare cells is enhanced in precision, a degree of misclassification of major cell types may arise.

The Sankey plot of MarsGT, GiniClust, SCMER and RaceID on a simulation dataset. The left legend of each Sankey plot is the true label of the simulation dataset, and the right legend of each Sankey plot is the predicted label of corresponding tool.

9 It is difficult to translate the metrics of NMI, Purity and Entropy into real-world performance. In addition, it could be more useful to have something simple like the adjusted Rand index against the original labels.

Response: Thanks for your suggestion. We did not include the ARI index for evaluation because it may introduce bias when evaluating rare cell populations [1]. ARI should be used when the reference clustering has large equal-sized clusters [2]. To answer your question here, we performed an ARI comparison, as shown below. The result shows that MarsGT still has the best performance in terms of ARI.

Rare cell population identification on Rare1_Major4, Rare1_Major9 and Rare5_Major10 datasets benchmarking with clustering-like tools evaluated via ARI.

NMI, Purity, and Entropy are important indexes that have been previously used for rare cell evaluation [3]. All evaluation indexes require true labels.

For Purity, it reflects the purity in each cluster. In the following illustration, there are three clusters, and we consider the major element in each cluster to be consistent with the true labels. The major element in cluster 1 is fork with five number, the major element in cluster 2 is circle with four numbers, the major element in cluster 3 is rhombus with three numbers. The Purity is calculated as follows:

$$\text{Purity} = \frac{1}{17} \times (5 + 4 + 3) \approx 0.7059$$

It is not hard to see the closer the purity is to 1, the better the clustering result. If all elements are the same in each cluster and consistent with the true label, then the Purity

equals 1.

For NMI, $NMI(P, T) = \frac{I(S, T)}{\min[H(S), H(T)]}$, $S = \{s_1, s_2, \dots, s_s\}$ be the set of predicted clusters, and $T = \{t_1, t_2, \dots, t_T\}$ be the set of true labels.

If S and T are independent of each other, and the prediction label is completely random for the real label, then the numerator is 0 and the NMI is 0. If S and T are exactly the same, then the numerator is 1 and the denominator is 1, then NMI is 1.

For entropy, it reflects how chaotic a cluster prediction result is compared to the true label. Lower entropy means higher clustering accuracy.

[1] Shengquan C, Boheng Z, Xiaoyang C, Xuegong Z, Rui J. stPlus: a reference-based method for the accurate enhancement of spatial transcriptomics. *Bioinformatics*, 37, i299-i307.

[2] Romano S. et al. Adjusting for chance clustering comparison measures. *J. Mach. Learn. Res.*, 17, 4635–4666.

[3] Schwartz, G. W. et al. TooManyCells identifies and visualizes relationships of single-cell clades. *Nature Methods* 17, 405-+.

10 A statement such as “MarsGT was 5.66% higher than the NMI score of the existing best-performing tool on the independent test dataset” is difficult to decipher. What does 5.66% translate to in practice? The results could be better demonstrated with specific graphical illustrations of simulated data experiments.

Response: We apologize for the inaccurate statement. We change the sentence to “*MarsGT delivered the best performance among all rare cell identification tools, achieving an NMI score that was 7.14% higher than that of the second-best-performing tool (GiniClust) in the independent test dataset (PBMC-test).*” The 7.14% was calculated by:

$$(0.7924 - 0.7358) / 0.7924 = 0.0566 / 0.7924 = 7.14\%$$

We add a graphical illustration here to answer your question.

11. When attention turns to more real-world data settings, e.g. in Figure 2D, it is difficult to understand the overall clustering performance because every method groups the cells in different ways. MarsGT appears more sensitive to the rare populations but the UMAP shows it does various things to the larger cell populations (merges cell types, or splits cell types). What is the trade-off? How is this controlled in MarsGT?

Response: Thank you for your insightful questions. In Fig. 2d, based on the benchmark label, there are two rare cell populations comprising less than 1% of cell numbers, and six rare cell populations with less than 3% of cell numbers.

The tools available to identify rare cell types can be broadly categorized into two main groups based on their recognition results. The first category includes tools that can only distinguish between rare and non-rare cell types, such as CellSIUS, FIRE, and GapClust. (1) CellSIUS is designed to identify multiple rare cell populations. We observed that CellSIUS could identify three populations with less than 3% rare cells; however, the rare cells identified differed from the benchmark, while also misidentifying part of a non-rare cell type as rare cells. (2) FIRE is designed to identify only one rare cell population. The results showed FIRE has a gap with the actual label results. (3) GapClust is designed to identify only one rare cell population. The results showed GapClust failed to identify any rare cell population.

The second category includes tools that can identify both rare and major cell populations simultaneously, such as GiniClust, RaceID, and SCMER. These tools cannot accurately identify rare cell populations and tend to merge and split major cell types. The performance of other tools demonstrates that existing tools do not accurately identify rare cell types. Simultaneously, the results also show that stratifying both rare cell populations and major cell populations presents a significant challenge. MarsGT demonstrated the best performance in identifying rare cell populations and also performed better than other tools in identifying major cell populations.

As shown in Figure 2D, although we have showcased superior performance in recognizing both rare and major cell types, we still merged B1 B and Naïve CD20+ B into

a single cell population, merged HSC and G/Mprog into a single cell population, and split CD14+ Mono into two cell populations in the major scenario. Our method does not fully restore the benchmark results because some cell types are difficult to distinguish. For instance, both B1 B and Naïve CD20+ B belong to B cells. For CD14+ Mono, there may be two subtypes as identified by MarsGT.

Our method primarily focuses on the identification of rare cell populations, and we have designed specific components to achieve this aim, as Reviewer #2, Question 4 mentioned (Please check the response to Reviewer # 1 Question 1). As you observed, in order to ensure the accuracy in rare cell identification, we did sacrifice some degree of performance in identifying major cell populations. Details can be found in the response to Reviewer #2, Question 8.

12. Further, how were the ground truth labels derived? Is there uncertainty in these labels?

Response: We appreciate this critical question. First, we admit that, regardless of the dataset, uncertainty in cell labels always exists. The benchmarking datasets used in MarsGT originated from the NeurIPS competition (<https://openproblems.bio/neurips>). In the original paper [1], the authors combined machine learning methods with an expert committee of scientists to annotate the cell labels. Raw data was first processed, annotated, and formatted to be usable by machine learning methods. As standards in single-cell analysis are rapidly evolving, the authors leveraged their previous work identifying best practices. They convened an expert committee of scientists from Helmholtz, Yale, Chan Zuckerberg Biohub, VIB–Ghent University, and Cellarity and consulted additional experts from Helmholtz Center Munich, Harvard, the Sanger Institute, and Stanford University to assist with cell annotation. The result of this effort is a high-quality, fit-for-purpose benchmark dataset for multi-modal single-cell analysis.

[1] Luecken, Malte D., et al. "A sandbox for prediction and integration of DNA, RNA, and proteins in single cells." Thirty-fifth conference on neural information processing systems datasets and benchmarks track (Round 2). 2021.

13. "Moreover, 12 rare cellular populations were distinguished, eight of which boast a 95% confidence level as highlighted by scPower". What does this statement mean? What does the 95% confidence level from scPower refer to?

Response: We meant to say MarsGT identified 12 clusters as rare cell populations, yet only eight of these clusters achieved statistical significance according to an independent tool, scPower. scPower is a statistical method used to define the necessary sample size for research or an experiment, aiming to detect an effect of a specific size while ensuring a desired significance level and power.

A 95% confidence level implies a 0.95 probability that a test will reject the null hypothesis when the alternative hypothesis is true, essentially indicating a 0.05 risk of committing a Type I error. In our context, this 95% confidence level suggests a 95% certainty that the detected rare cell population is genuinely rare and not a result of false positives. In our manuscript, to ensure a robust analysis, we have concentrated on rare cell populations with 95% confidence and have opted not to analyze those rare cell

populations that did not achieve statistical significance (Please also check the response to Reviewer #1 Question 2).

Overall, the paper presents potentially interesting methodology but from my personal perspective, the methodology description needs substantial revision to better explain and justify the design choices and most importantly provide the average reader with a more intuitive understanding of how it works. The experiments section is substantial and great efforts have been placed here but the simulated (controlled) experiment setting is the only place where the authors have full control. This section needs to be refined both in terms of description but also in the design of the experiments to match them with real world data.

Response: Thank you for your comments and constructive suggestions. We have elaborated on the method in greater detail, explaining the rationale behind each step to facilitate a clearer understanding for readers of our processes, design choices, and objectives. Previously, the simulation datasets we utilized were simplistic; hence, we have introduced 150 new, more complex simulation datasets. By increasing the number of cell types and the count of cells within each dataset, we aim to ensure that these simulations more closely mirror real data. We organized these into three categories, with each category comprising 50 simulation datasets. Notably, MarsGT demonstrated superior performance on both the original, simpler simulation sets and the newly introduced, more complex datasets.

Reviewer #1 (Remarks to the Author):

The authors addressed all my questions.

Reviewer #2 (Remarks to the Author):

The authors have generally responded well to all my previous comments. However, the insight provided by the Sankey plots in the author response should be included as a discussion point in the main manuscript and illustrated as a supplementary item. This is to ensure that potential future users are aware of the trade-offs.

Department of Biomedical Informatics

College of Medicine

3049 Pelotonia Research Center

2255 Kenny Road

Columbus, OH, 43210

Reviewer #1 (Remarks to the Author)

Reviewer #1 (Remarks to the Author):

The authors addressed all my questions.

Reviewer #2 (Remarks to the Author):

The authors have generally responded well to all my previous comments. However, the insight provided by the Sankey plots in the author response should be included as a discussion point in the main manuscript and illustrated as a supplementary item. This is to ensure that potential future users are aware of the trade-offs.

Response: We discussed the trade-off of potentially breaking up (or merging) other non-rare classes on Page 21, Lines 613-616 of the main manuscript, and added the corresponding Sankey Plot for the simulation datasets in Supplementary Figure 22.